# ACTION-ADAPTIVE CONTINUAL LEARNING: ENABLING POLICY GENERALIZATION UNDER DYNAMIC ACTION SPACES

## ABSTRACT

Continual Learning (CL) is a promising paradigm that enables agents to learn a sequence of tasks, accumulating knowledge learned in the past and using it for problem-solving or future task learning. However, existing CL methods often assume that the agent's capabilities remain static within dynamic environments, which doesn't reflect real-world scenarios where capabilities dynamically change. This paper introduces a new and realistic problem: *Continual Learning with Dynamic Capabilities* (CL-DC), posing a significant challenge for CL agents: How can policy generalization across different action spaces be achieved? Inspired by the cortical functions, we propose an **A**ction-**A**daptive **C**ontinual **L**earning framework (AACL) to address this challenge. Our framework decouples the agent's policy from the specific action space by building an action representation space. For a new action space, the encoder-decoder of action representations is adaptively fine-tuned to maintain a balance between stability and plasticity. Furthermore, we release a benchmark based on three environments to validate the effectiveness of methods for CL-DC. Experimental results demonstrate that our framework outperforms popular methods by generalizing the policy across action spaces. [1]

## 1 INTRODUCTION

Continual Learning (CL, a.k.a. lifelong learning) is an emerging research field that aims to emulate the human capacity for lifelong learning and tackles the challenges of long-term, real-world applications characterized by diversity and non-stationarity (Rolnick et al., 2019; Kessler et al., 2022). Specifically, CL in Reinforcement Learning (RL) extends traditional Deep Reinforcement Learning (DRL) by empowering agents with the ability to learn from a sequence of tasks, preserving knowledge from previous tasks, and using this knowledge to enhance learning efficiency and performance on future tasks. Although related CL works re-

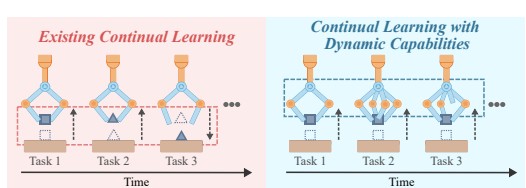

Figure 1: An example of two CL problems. **Existing CL**: A robot uses two fingers to grasp objects while the objects or grasping way changes. **CL-DC**: A robot initially trained with two fingers is upgraded to four fingers or loses a finger but must continue grasping objects.

quire the agent's ability to adapt to dynamic environments (Khetarpal et al., 2022), they typically assume that the agent's capabilities (action space) remain static while the external environment changes. This assumption does not reflect realistic situations where an agent's action space may dynamically change. Living systems not only need to adapt to radical changes in the environment (Emmons-Bell et al., 2019), but also need to deal with changes to their structure and function (Blackiston et al., 2015). Similarly, CL agents also need to deal with their dynamic capabilities (Kudithipudi et al., 2022). For example, the action space of agents in real-world applications may change due to software or hardware updates (Wang et al., 2019; Ding et al., 2023) or damages (Kriegman et al., 2019; Kwiatkowski & Lipson, 2019). Therefore, CL with the changes of action space is crucial for developing more sophisticated and adaptable artificial intelligence systems.

---

[1]Code and videos are released in Supplementary Material.

While existing works in RL (Chandak et al., 2020; Ding et al., 2023) have made initial explorations into the challenges posed by changing action spaces, these studies have certain limitations. They primarily focus on continual adaptation without addressing other critical issues in CL, such as catastrophic forgetting. Additionally, they only consider expanding action spaces, neglecting other types of changes in the action space. Building on these foundational studies, we propose a new and more general problem called *Continual Learning with Dynamic Capabilities* (CL-DC), where the agent needs to learning continually with different action spaces. Figure 1 illustrates the difference between CL-DC and existing CL. While existing CL requires exploring how to respond to environmental changes, CL-DC needs to maintain

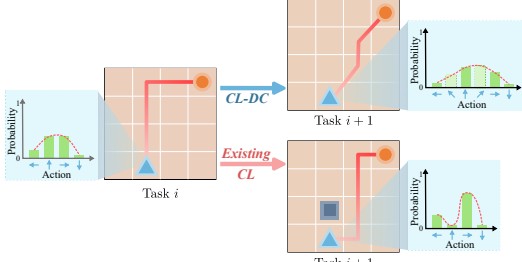

Figure 2: Different challenges of two problems. **Existing CL**: After the environment changes, the number of actions remains constant, while the probability distribution shifts (trend of the red line). **CL-DC**: After the action space changes, the number of actions changes, while the probability distribution is relatively stable.

the performance with changing action spaces, considering catastrophic forgetting and knowledge transfer. CL-DC supplements existing CL research by considering dynamics in a broader context. As an early step, it focuses on discrete action spaces and assumes that the task logic remains unchanged.

As shown in Figure 2, the main challenge of CL-DC is different from existing CL. The main challenge of existing CL is dealing with the significant shift of the probability distribution of the actions after the environment changes, while the main challenge of CL-DC is to cope with changes in the actions' number after the action space changes. Although a general policy can be obtained using the union of all action spaces, the union optimum may not an optimum for all specific action space. In addition, this requires prior knowledge about all action spaces, which is not always sufficient. In summary, CL-DC can be formally modeled as the following problem: *How to achieve policy generalization across different action spaces with the same task logic?*

Animals, including humans, consistently perform behaviors even years after learning (Emmons-Bell et al., 2019; Blackiston et al., 2015). It is due to the brain's ability to represent actions in a latent space, allowing for the generalization across different contexts. Precisely, the stability of latent dynamics of neural activity reflects a fundamental feature of learned cortical function, leading to stable and consistent behavior (Gallego et al., 2020). In addition, the research on self-supervised learning (SSL) for RL has been shown to be effective in improving the generalization ability of the agent (Liu et al., 2024; Fang & Stachenfeld, 2024).

Inspired by these, we propose an **A**ction-**A**daptive **C**ontinual **L**earning framework (AACL) to address the challenge of CL-DC. AACL first builds an action representation space by learning an encoder-decoder. It is trained through SSL on transitions collected from the agent's exploration of the environment. The encoder maps the agent's actions to action representations, and the decoder maps them to action probabilities. Once trained, the encoder-decoder is fixed, and the agent's policy is trained based on the action representation space. When the action space changes, the decoder's structure is adaptively updated to accommodate the size of the new action space. The agent then explores the environment with the new action space and adaptively fine-tunes the encoder-decoder, where the parameters update of the decoder are constrained to enhance stability. In this process, the function of the decoder is similar to that of the cerebellum of humans, while the policy corresponds to that of the primary motor cortex. The former is essential for learning new mapping (plasticity), but the latter is vital for consolidating the new mapping (long-term retention/stability) (Haar et al., 2015; Gazzaniga et al., 2019; Weightman et al., 2023).

To evaluate the performance of CL methods in CL-DC, we release a benchmark based on three environments, which includes three sets of tasks with different discrete action spaces and three task sequence situations (Expansion, Contraction, and their combinations). Experimental results demonstrate that AACL effectively handles CL-DC compared to other methods.

Our contributions can be summarized as follows:

- We formally propose the *Continual Learning with Dynamic Capabilities* problem (CL-DC), supplementing the existing CL by focusing on changing action spaces.

- We propose a CL framework called AACL, which decouples the policy from the specific action space by building an action representation space. By adaptively fine-tuning the networks, this framework maintains a good balance between stability and plasticity. As the first step in handling CL-DC, AACL provides valuable insights for further enhancing the generalization ability of CL agents.

- We release a benchmark of CL-DC to evaluate the performance of popular CL methods. Extensive experimental results on three environments and three situations show the distinct challenge of CL-DC and demonstrate that AACL is more effective than others.

## 2 RELATED WORKS

Continual learning agent focuses on learn multiple tasks sequentially without prior knowledge, generating significant interest due to its relevance to real-world artificial intelligence applications (De Lange et al., 2022; Wang et al., 2024a). Related works are also known as continual reinforcement learning (CRL) and is a subfield of CL (Khetarpal et al., 2022; Abel et al., 2023).

A central issue in CRL is catastrophic forgetting, which has led to various strategies for knowledge retention. PackNet and related methods (Mallya & Lazebnik, 2018; Schwarz et al., 2021; Ben-Iwhiwhu et al., 2023) preserve model parameters but often require knowledge of task count. Experience replay techniques such as CLEAR (Rolnick et al., 2019) and CPPO (Zhang et al., 2024) use buffers to retain past experiences but face memory scalability challenges. In addition, some methods prevent forgetting by maintaining multiple policies or a subspace of policies (Schöpf et al., 2022; Gaya et al., 2022). Furthermore, task-agnostic CRL research indicates that rapid adaptation can also help prevent forgetting (Caccia et al., 2023; Dick et al., 2024).

Another issue in CRL is transfer learning, which is crucial for efficient policy adaptation. Naive approaches, like fine-tuning, train a single model on each new task and provide good scalability and transferability but suffer from forgetting (Gaya et al., 2022). Regularization-based methods, such as EWC (Kirkpatrick et al., 2017; Wang et al., 2024a), have been proposed to prevent this side effect, but often reduce plasticity (Lomonaco et al., 2020; Wang et al., 2020). Some architectural innovations have been proposed to balance the trade-off between plasticity and stability (Rusu et al., 2016; Berseth et al., 2022). Furthermore, methods like OWL (Kessler et al., 2022) and MAXQINIT (Abel et al., 2018) leverage policy factorization and value function transfer, respectively, for efficient transfer learning.

Most existing CRL methods perform well when applied to sequences of tasks with a static action space and a dynamic environment, such as when environmental parameters are altered, or the objectives within the same environment are different (Pan et al., 2025). However, their effectiveness is greatly diminished when the agent's action space dynamically change. Our proposed framework aims to overcome this limitation by building an action representation space. [More related works of SSL and RL are provided in Appendix B.]

## 3 CL WITH DYNAMIC CAPABILITIES

### 3.1 PRELIMINARIES

The learning process of an agent can be formulated as a Markov Decision Process (MDP) $\{\mathcal{S}, \mathcal{A}, \mathcal{P}, \mathcal{R}\}$, which is commonly used in RL. A MDP represents a problem instance that an agent needs to solve over its lifetime. Here, $\mathcal{S}$ and $\mathcal{A}$ denote the state and action space, respectively, while $\mathcal{P} : \mathcal{S} \times \mathcal{S} \times \mathcal{A} \to [0, 1]$ is the transition probability function, and $\mathcal{R} : \mathcal{S} \times \mathcal{A} \to [r^{\min}, r^{\max}]$ is the reward function. At each time step, the learning agent perceives the current state $S_t \in \mathcal{S}$ and selects an action $A_t \in \mathcal{A}$ according to its policy $\pi : \mathcal{S} \times \mathcal{A} \to [0, 1]$. The agent then transitions to the next state $S_{t+1} \sim \mathcal{P}(\cdot|S_t, A_t)$ and receives a reward $R_t = \mathcal{R}(S_t, A_t, S_{t+1})$. The value function for policy $\pi$ is defined as $V^\pi(s) = \mathbb{E}_\pi \left[ \sum_{j=0}^{H-t} \gamma^j R_{t+j} | S_t = s \right]$, where $\gamma$ is the discount factor of the reward, and $H$ is the horizon. The goal of an agent is to find an optimal policy $\pi^*$ to maximize the expected return $\mathbb{E}_{\pi^*} \left[ \sum_{t=0}^{H} \gamma^t \mathcal{R}(S_t, A_t, S_{t+1}) \right]$, which is the value function of the initial state.

## 3.2 PROBLEM FORMALIZATION

In the real world, the capabilities of an agent may change over time. To explore this, we introduce a new problem called *Continual Learning with Dynamic Capabilities (CL-DC)*. This problem can be formally defined as a sequence of MDPs $\{(\mathcal{S}, \mathcal{A}^i, \mathcal{P}^i, \mathcal{R}^i) | i = 1, 2, ..., N\}$, where $N$ is the total number of MDPs and $\mathcal{A}^i$ represents the action space available to the agent at MDP $i$. Following the convention in CL, we still use "task" to represent each MDP in the sequence. Each task in the sequence shares a common state space, but differs in the action space and implicitly in the transition probability function $\mathcal{P}^i$ and reward function $\mathcal{R}^i$, which is influenced by the action space $\mathcal{A}^i$.

To simplify the problem, we assume that the action space is discrete and finite, and we focus on the impact of changing action spaces on the learning process while assuming $\mathcal{P}^i$ and $\mathcal{R}^i$ remains "conceptually similar" across tasks. The latter assumption means for a same action in different action spaces, the transition probability and reward value are the same. This allows the task logic to be the same after the action space is changed. [More details are provided in Appendix E.]

Then, the dynamics can be characterized by differences in successive action spaces. For each task $i > 1$, the action space $\mathcal{A}^i$ can be related to the previous action space $\mathcal{A}^{i-1}$ in one of the following situations ($\mathcal{A}^i \neq \mathcal{A}^{i-1}$):

1. **Expansion**: $\mathcal{A}^{i-1} \subset \mathcal{A}^i$ (new actions are added).
2. **Contraction**: $\mathcal{A}^{i-1} \supset \mathcal{A}^i$ (some actions are removed).
3. **Partial Change**: $\mathcal{A}^{i-1} \cap \mathcal{A}^i \neq \emptyset$ and $\mathcal{A}^{i-1} \nsubseteq \mathcal{A}^i$ and $\mathcal{A}^i \nsubseteq \mathcal{A}^{i-1}$ (some actions are removed, and some actions are added).
4. **Complete Change**: $\mathcal{A}^{i-1} \cap \mathcal{A}^i = \emptyset$ (all actions are removed, and new actions are added).

Previous work focuses on the first situation from the perspective of transfer reinforcement learning (Chandak et al., 2020; Ding et al., 2023). However, the other situations are less explored in the literature, especially in the broader context of CL. In this work, we take a step further to address the problem of CL-DC by considering the first two situations and their combinations.

The policy of the agent on task $i$ is denoted as $\pi_{\theta^i} : \mathcal{S} \times \mathcal{A}^i \rightarrow [0, 1]$, where $\theta^i$ represents the policy parameters. After learning on tasks $\{1, 2, \cdots, i\}$, the agent's objective is to learn a policy that maximizes the average expected return overall tasks. This can be formally expressed as:

$$\max_{\theta^i} \frac{1}{i} \sum_{j=1}^{i} \mathbb{E}_{\pi_{\theta^i}} \left[ \sum_{t=0}^{H^j} \gamma^t \mathcal{R}(S_t, A_t^j, S_{t+1}) \right], \tag{1}$$

where $H^j$ is the horizon of task $j$, and $A_t^j \in \mathcal{A}^j$ is the action at the $t$-th step on task $j$. The expected return at each task is related to the current policy $\pi_{\theta^i}$ and the corresponding action space $\mathcal{A}^j$.

# 4 ACTION-ADAPTIVE CONTINUAL LEARNING

## 4.1 FRAMEWORK

Our goal is to design a framework for generalized policy learning that can adapt to the changing action space, enhancing agent adaptation to dynamic capabilities. Recent neuroscience research has shown that the stability of latent dynamics of neural activity reflects a fundamental feature of learned cortical function that leads to long-term and consistent human behavior (Gallego et al., 2020). Furthermore, research on SSL for RL has demonstrated its effectiveness in improving the agent's generalization ability (Chandak et al., 2019; Liu et al., 2024; Fang & Stachenfeld, 2024). Drawing inspiration from these findings, we propose a new framework, named **A**ction-**A**daptive **C**ontinual **L**earning (AACL), to enables the agent to generalize policy across different action spaces.

Figure 3 illustrates the overview of AACL. By decoupling the policy of the agent from the action space, the policy can be generalized to new action spaces efficiently. The interaction between the agent and the environment at each task is achieved through the action representation space. Each task in AACL consists of a two-stage process: **Exploration Stage)** As shown in the left part of Figure 3, the agent explores the current action space in the environment, collects the transitions (state-action-state pairs), and learns an encoder-decoder through SSL. The encoder maps the action space

to an action representation space, and the decoder maps the action representation space to the action space. When the action space changes, the agent can adapt by adjusting the decoder's structure, while the parameters of the encoder and the decoder are updated. Furthermore, to enhance the stability of the policy, the update of the decoder's parameters is constrained by the previous policy. **Learning Stage)** As shown in the right part of Figure 3, the agent learns a policy based on the learned action representation space, rather than the specific action space of each task. Specifically, the action representation space is treated as the action space, and a standard RL policy is used to maximize the expected return. Once the action space changes, the agent needs to use the updated decoder. In this way, the policy can maintain stability through the action representation space, and the agent can adapt to the new action space efficiently.

## 4.2 ACTION REPRESENTATION SPACE BUILDING

We use SSL to build an action representation space. The agent explores the action space in the environment of task $i$ before learning the policy. It collects the transitions $\mathcal{T}^i = \{(s, a, s')_m | m = 1, 2, \cdots, M\}$ without reward, where $s'$ is the next state of $s$ after taking action $a$ and $M$ is the number of transitions. Based on the work in RL (Chandak et al., 2019; Fang & Stachenfeld, 2024), we believe that the features of the actions can be naturally represented by their influences of state changes. Therefore, the auxiliary task of the action representation is to predict the action $a$ given the current state $s$ and the next state $s'$. Specifically, for a transition $(s, a, s')$, the encoder $f_{\phi^i}$ parameterized by $\phi^i$ maps an action $a$ to an action representation $e \in \mathcal{E}$. The decoder $g_{\delta^i}^i$ parameterized by $\delta^i$ maps the action representation $e \in \mathcal{E}$ to the action probability. The processes are formulated as:

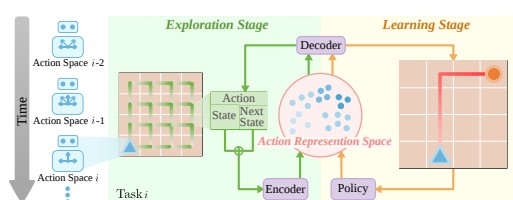

Figure 3: The overview of AACL. The tasks with different action spaces are learned sequentially. Each task consists of two stages: the exploration stage (**green**) and the learning stage (**yellow**). The former aims to build an action representation space, and the latter aims to learn a policy based on the learned space. This framework separates the policy from the action space, allowing the agent's generalization across different action spaces.

$$
\begin{aligned}
\textbf{Encoding} &: e = f_{\phi^i}(s, s'), \forall s \in \mathcal{S}, \forall s' \in \mathcal{S}, \\
\textbf{Decoding} &: a \sim g_{\delta^i}^i(\cdot | e), \forall e \in \mathcal{E}.
\end{aligned}
\tag{2}
$$

Although we use different superscripts to represent decoders in different tasks for clarity, the structure is updated rather than being task-specific. Therefore, $g_{\delta^i}^i$ needs to map action representation $e$ to the action probability of any action space from past and current tasks during testing.

The probability of an action $a$ given the state $s$ and the next state $s'$ can be represented as $g_{\delta^i}^i(a | f_{\phi^i}(s, s'))$. To measure the difference between the true action probability and the predicted action probability, we use the cross-entropy loss as the loss function of the encoder-decoder network:

$$
\begin{aligned}
\mathcal{L}(\phi^i, \delta^i) &= - \sum_{(s,a,s') \in \mathcal{T}} \log P(a | s, s') \\
&= - \sum_{(s,a,s') \in \mathcal{T}} \log g_{\delta^i}^i(a | f_{\phi^i}(s, s')).
\end{aligned}
\tag{3}
$$

This loss function only depends on the environmental dynamic data which is reward-agnostic, the agent can build the action representation space $\mathcal{E}$ with low computational cost.

After the SSL process, the agent can use the learned action representation space to interact with the environment. The original policy $\pi^i : \mathcal{S} \times \mathcal{A} \to [0, 1]$ can be represented by another policy $\tilde{\pi}_{\theta^i} : \mathcal{S} \to \mathcal{E}$ and the decoder $g_{\delta^i}^i : \mathcal{E} \times \bigcup_{j=1}^i \mathcal{A}^j \to [0, 1]$:

$$
\pi^i(a | s) = g_{\delta^i}^i(a | \tilde{\pi}_{\theta^i}(s)).
\tag{4}
$$

Then the policy can be trained by a standard RL algorithm to maximize the expected return:

$$J(\theta^i) = \mathbb{E}_{\tilde{\pi}_{\theta^i}} \left[ \sum_{t=0}^{H^i} \gamma^t R(S_t, A_t, S_{t+1}) \right]. \tag{5}$$

### 4.3 ACTION-ADAPTIVE FINE-TUNING

In the new task $i + 1$, the policy needs to generalize to the new action space $\mathcal{A}^{i+1}$. The structure of the action representation decoder $g_{\delta^{i+1}}^{i+1}$ needs to be updated to adapt to the new action space. If the action space is expanded, that is $\mathcal{A}^{i+1} \supset \mathcal{A}^i$, the network is expanded by adding new neurons. The parameters of the old neurons are fixed and the parameters of new neurons are initialized randomly. If the action space is contracted, that is $\mathcal{A}^{i+1} \subset \mathcal{A}^i$, the output corresponding to the actions that are not in the new action space is masked. This strategy has been studied in the works of architecture-based CL methods (Rusu et al., 2016; Mallya & Lazebnik, 2018; Mallya et al., 2018).

During training on the transitions of the new task, the decoder is fine-tuned with constraints. We use the elastic weight consolidation to constrain the fine-tuning process, as it has been shown to be effective in mitigating the catastrophic forgetting in CL (Kirkpatrick et al., 2017). The encoder is also fine-tuned in the new tasks to continuously refine the action representation space $\mathcal{E}$. We do not impose constraints on the encoder because we have found that its plasticity is crucial for learning a good representation space. The loss function of the decoder network in Equation 3 is modified to include the regularization term:

$$\begin{aligned}
\mathcal{L}(\phi^{i+1}, \delta^{i+1}) = &- \sum_{(s,a,s') \in \mathcal{T}^{i+1}} \log g_{\delta^{i+1}}^{i+1}(a | f_{\phi^{i+1}}(s, s')) \\
&+ \frac{\lambda}{2} \sum_{j=1}^{i} \sum_{k} F_k^j \left( \delta_k^{i+1} - \delta_k^j \right)^2,
\end{aligned} \tag{6}$$

where $F_k^j$ is the $k$-th diagonal element of the Fisher information matrix of the parameters of the decoder network on task $j$, and $\lambda$ is a regularization coefficient to balance the two terms. After the fine-tuning process, the agent can use the new decoder to interact with the environment. In order to maintain consistency with the standard pipeline of CL, we still update the policy in the new action space. This process does not use any regularization, and the objective function is the same as Equation 5. [The algorithm is provided in Appendix A.]

## 5 EXPERIMENTS

### 5.1 BENCHMARK

To evaluate the performance of CL methods in CL-DC, we establish a benchmark with changing action spaces. This benchmark comprises sequences with tasks that share identical state, reward, and transition dynamics but possess different action spaces. [Detailed descriptions of experimental settings, network structures, hyperparameters, and the metrics are provided in Appendix C.]

**Environments.** The environments of our tasks are based on MiniGrid (Chevalier-Boisvert et al., 2023), Procgen (Cobbe et al., 2020) and Arcade Learning Environment (ALE) (Bellemare et al., 2013). These environments feature image-based observations, a discrete set of possible actions. For better demonstration of the impact of action space changes, we use Bigfish of Procgen and Atlantis of ALE in our experiments. The agents in these environments can move in different directions. To simulate the dynamic action space, we introduce some additional actions or remove some existing actions. Then, we design three tasks with different numbers of actions. When the agent switches to a new task, the set of available actions may either increase or decrease. The agent can only observe the action space of the current task.

**Compared Methods.** We select three types of CL methods in RL to compare with CL-DC: one *replay-based* method, **CLEAR** (Rolnick et al., 2019); two *regularization-based* methods, **EWC** (Kirkpatrick et al., 2017) and **online-EWC** (Schwarz et al., 2018); and one *architecture-based* method, **Mask** (Ben-Iwhiwhu et al., 2023). Additionally, we take the DRL methods trained with fine-tuning (named **FT**) and independently (named **IND**) across tasks as baselines. In the implementation of these methods, we adapt them to CL-DC by using the largest action space of all tasks.

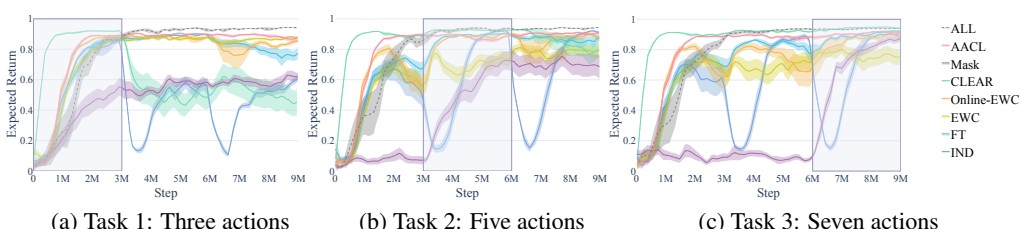

(a) Task 1: Three actions      (b) Task 2: Five actions      (c) Task 3: Seven actions

Figure 4: Performance of eight methods on three **MiniGrid** tasks in the **expansion** situation.

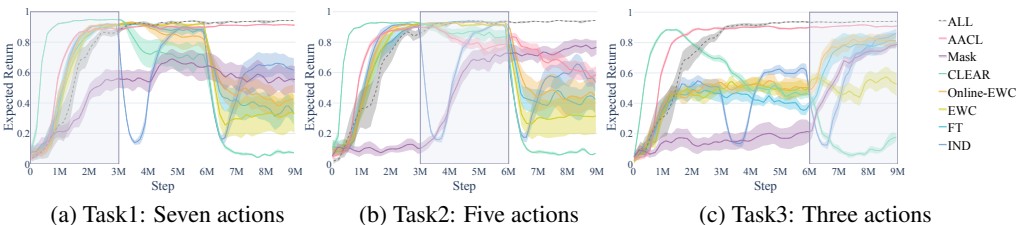

(a) Task1: Seven actions      (b) Task2: Five actions      (c) Task3: Three actions

Figure 5: Performance of eight methods on three **MiniGrid** tasks in the **contraction** situation.

This adaptation necessitates prior knowledge, while our framework does not require it. In order to better understand the challenge of CL-DC, we also introduce a baseline that is always able to access all action spaces (named **ALL**). It does not involve CL and its final performance can be regarded as an upper bound of other methods. We use IMPALA (Espeholt et al., 2018) as the underlying RL algorithm for all methods to ensure a fair comparison.

**Metrics.** To evaluation in CL-DC, we use the expected return to measure the performance of agents. Following the standard practice of CL (Díaz-Rodríguez et al., 2018; Wolczyk et al., 2021; Li et al., 2024b), we use three metrics based on the agent's performance throughout different phases of its training process: **continual return**, **forgetting**, and **forward transfer** (Powers et al., 2022). The continual return is the average performance achieved by the agent on all tasks after completing all training, which is consistent with the agent's objective in Equation 1. The forgetting compares the expected return achieved for the earlier task before and after training on a new task, while the forward transfer compares the expected return achieved for the later task before and after training on an earlier task. Furthermore, the forward transfer also measures the zero-shot generalization ability.

### 5.2 COMPETITIVE EXPERIMENTS

To expeditiously evaluate the effectiveness of our framework, we first compare it with other methods on MiniGrid tasks. Each task is trained for 3M steps and replicated with 10 random seeds to ensure statistical reliability. During each task's training phase, the agent is not only trained on the current task but also **periodically evaluated on all tasks**, including tasks it has previously encountered. The results presented in the evaluation plots and the total evaluation metric reported in the table below were computed as the mean of runs per method, with the shaded area and errors denoting the 95% confidence interval. Each subplot in the evaluation plots depicts the expected return of the agent evaluated on the corresponding task during training on all tasks, with the x-axis representing the total number of training steps across all tasks. The blue-shaded rectangular area indicates the **training phase** of the current task. We employ exponential moving averages to smooth the results for better visualization. As tasks are learned independently of other tasks in IND, there is no notion of forward transfer. [Further details and more experiments (combined situations, longer sequences, and hyperparameter sensitivity analysis, etc) are provided in Appendix D]

**Overall Performance.** Figures 4 and 5 show the evaluated performance of eight methods on Mini-Grid tasks with action spaces that are either expanding or contracting, respectively. The return curve of AACL (red line) is generally higher than that of other methods across all tasks, suggesting that AACL adapts more effectively to changing action spaces and achieves superior performance. In the **expansion** situation, the overall performance of some methods slightly improves as training progresses (more evident in Figure 15 and Table 9). This phenomenon suggests that an expanding action space may facilitate policy generalization, echoing the principle of curriculum learning (Wang

Table 1: Continual learning metrics of eight methods and three variants of AACL across three **MiniGrid** tasks in situations of **expansion** and **contraction**. The average continual return of ALL is 0.94, which is not provided in the table. Continual return and forward transfer are abbreviated as "Return" and "Transfer", respectively. The top three results of compared methods are highlighted in green, and the depth of the color indicates the ranking.

| Methods | Expansion | | | Contraction | | |
|---|---|---|---|---|---|---|
| | Return↑ | Forgetting↓ | Transfer↑ | Return↑ | Forgetting↓ | Transfer↑ |
| IND | $0.81 \pm 0.02$ | $0.08 \pm 0.02$ | – | $0.67 \pm 0.06$ | $0.26 \pm 0.05$ | – |
| FT | $0.86 \pm 0.03$ | $0.03 \pm 0.02$ | $0.48 \pm 0.03$ | $0.52 \pm 0.07$ | $0.39 \pm 0.06$ | $0.39 \pm 0.03$ |
| EWC | $0.81 \pm 0.04$ | $-0.04 \pm 0.01$ | $0.43 \pm 0.05$ | $0.39 \pm 0.11$ | $0.40 \pm 0.09$ | $0.47 \pm 0.03$ |
| online-EWC | $0.87 \pm 0.02$ | $0.02 \pm 0.01$ | $0.40 \pm 0.05$ | $0.56 \pm 0.09$ | $0.34 \pm 0.06$ | $0.44 \pm 0.03$ |
| Mask | $0.72 \pm 0.05$ | $-0.04 \pm 0.04$ | $0.02 \pm 0.03$ | $0.70 \pm 0.06$ | $-0.02 \pm 0.04$ | $0.08 \pm 0.03$ |
| CLEAR | $0.73 \pm 0.06$ | $0.21 \pm 0.06$ | $0.58 \pm 0.01$ | $0.11 \pm 0.02$ | $0.58 \pm 0.03$ | $0.46 \pm 0.02$ |
| AACL | $0.90 \pm 0.01$ | $-0.02 \pm 0.01$ | $0.57 \pm 0.02$ | $0.80 \pm 0.03$ | $0.04 \pm 0.03$ | $0.60 \pm 0.01$ |
| AACL-O | $0.86 \pm 0.03$ | $0.01 \pm 0.02$ | $0.52 \pm 0.02$ | $0.73 \pm 0.06$ | $0.06 \pm 0.03$ | $0.60 \pm 0.01$ |
| AACL-E | $0.89 \pm 0.03$ | $-0.03 \pm 0.03$ | $0.55 \pm 0.04$ | $0.76 \pm 0.05$ | $0.13 \pm 0.04$ | $0.55 \pm 0.03$ |
| AACL-OE | $0.88 \pm 0.01$ | $0.02 \pm 0.01$ | $0.56 \pm 0.02$ | $0.71 \pm 0.05$ | $0.17 \pm 0.04$ | $0.46 \pm 0.02$ |

et al., 2022). However, some methods experience a performance drop after training task changes (e.g., step 3M in Figure 18b), highlighting the challenge of policy generalization across different action spaces. AACL, with relatively small performance fluctuations upon action space changes, demonstrates its superior generalizability. In the **contraction** situation, performance changes are more pronounced. Most methods suffer significant performance shifts when the action space is reduced, indicating that policies trained on larger action spaces may not transfer well to smaller ones. Although AACL sometimes experiences a larger performance drop compared to Mask, which focuses on mitigating catastrophic forgetting, it generally outperforms other methods. These findings serve as evidence that our framework is effective in addressing CL-DC.

**Continual Learning Performance.** Table 1 presents the evaluation results in terms of CL metrics. The forward transfer metric is particularly noteworthy, as it measures the agent's ability to leverage knowledge from previous tasks and indicates zero-shot generalization to new action spaces. AACL exhibits the highest forward transfer underscoring the benefit of the action representation space for generalization. The forgetting metric of all methods is relatively high in the contraction situation, further underscoring the policy generalizability challenge in CL-DC. When some actions are removed, the optimal policy may change significantly, leading to a performance drop. Note that regularization-based methods (EWC and online-EWC) can not mitigate catastrophic forgetting in this situation, possibly due to the large difference in networks' parameters between different action spaces. Although AACL does not achieve the best score for the forgetting metric, its exceptional forward transfer capabilities and strong average performance accentuate its proficiency.

## 5.3 Ablation Study

We conduct an ablation study on MiniGrid tasks to investigate what affects AACL's performance in CL-DC. We consider three variants: **AACL-O**, which omits the regularization during fine-tuning, **AACL-E**, which uses the same regularization for the encoder and decoder, and **AACL-OE**, which only uses the regularization for the encoder. The results, presented in Table 1, reveal that AACL-O exhibits better forward transfer than most methods, demonstrating that the action representation space is helpful for a more generalized policy. The comparison between AACL and AACL-O in forgetting and forward transfer suggests that regularization may improve policy stability. Nevertheless, AACL's superior continual return demonstrates that this balance is beneficial for the stability-plasticity trade-off essential in continual learning systems (Wang et al., 2024a). Compared with AACL, additional regularization in AACL-E damages the forward transfer. Moreover, the forgetting metrics of AACL-E and AACL-OE in the contraction situation are worse than AACL-O and AACL. These may indicate that the plasticity of the encoder is essential for the learning of the action representation space.

## 5.4 More Challenging Experiments

To further evaluate the effectiveness of AACL, we conduct experiments on the Bigfish and Atlantis tasks. These tasks are more challenging than MiniGrid tasks due to their larger state space and more complex control logic. The change of the action space within Atlantis will exert a more pronounced influence on the agent's performance compared to Bigfish, as each action is important to achieving

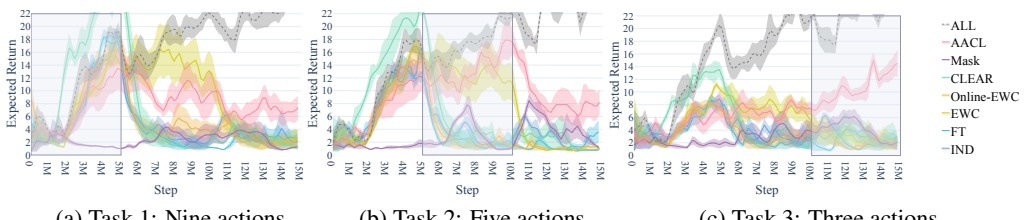

(a) Task 1: Nine actions        (b) Task 2: Five actions        (c) Task 3: Three actions

Figure 6: Performance of eight methods on three **Bigfish** tasks in the **contraction** situation.

Table 2: Continual learning metrics of eight methods across three **Bigfish** tasks or three **Altantis** tasks in the **contraction** situation. The average continual return of ALL in Bigfish is 24.77, and in Atlantis is 296, 949, which are not provided in the table. The top three results are highlighted in green, and the depth of the color indicates the ranking.

| Methods | Bigfish | | | Atlantis | | |
|---|---|---|---|---|---|---|
| | Return↑ | Forgetting↓ | Transfer↑ | Return↑ | Forgetting↓ | Transfer↑ |
| IND | $1.66 \pm 0.96$ | $0.18 \pm 0.02$ | – | $19,541 \pm 4,626$ | $0.32 \pm 0.05$ | – |
| FT | $3.01 \pm 1.39$ | $0.14 \pm 0.08$ | $0.23 \pm 0.02$ | $2,913 \pm 108$ | $0.52 \pm 0.01$ | $0.45 \pm 0.04$ |
| EWC | $1.74 \pm 1.04$ | $0.26 \pm 0.06$ | $0.16 \pm 0.06$ | $6,350 \pm 2,459$ | $0.48 \pm 0.04$ | $0.30 \pm 0.05$ |
| online-EWC | $1.84 \pm 0.80$ | $0.14 \pm 0.03$ | $0.18 \pm 0.07$ | $10,083 \pm 5760$ | $0.35 \pm 0.08$ | $0.23 \pm 0.03$ |
| Mask | $1.49 \pm 0.71$ | $-0.01 \pm 0.01$ | $0.06 \pm 0.06$ | $13,799 \pm 3,041$ | $0.08 \pm 0.05$ | $0.02 \pm 0.00$ |
| CLEAR | $1.48 \pm 0.44$ | $0.23 \pm 0.02$ | $0.11 \pm 0.04$ | $2,903 \pm 157$ | $0.59 \pm 0.01$ | $0.57 \pm 0.00$ |
| AACL | $10.03 \pm 1.94$ | $0.12 \pm 0.07$ | $0.19 \pm 0.05$ | $32,818 \pm 8,723$ | $0.45 \pm 0.08$ | $0.53 \pm 0.01$ |

the game's objective. Each experiment is trained for 15M (9M for Atlantis) steps and replicated with 5 random seeds to ensure statistical reliability. Other experimental configurations are consistent with those used in the MiniGrid experiments. As demonstrated in previous experiments, the contraction situation better highlights the challenges of CL-DC. Therefore, we focus on analyzing the results in this situation. [Please refer to the Appendix D.8 for other experimental results.]

Figures 6, 16 and Table 2 present the performance and metrics of eight methods across three tasks, respectively. The performance gap between ALL and other methods is more pronounced in these experiments, showing the challenges posed by the Bigfish and Altantis environments. Consistent with the MiniGrid experiments, AACL outperforms other methods across all tasks. All methods experience significant performance drops after training task changes, but AACL has a less volatile return curve due to its stability. Furthermore, AACL evidently improves performance during the training of the last task, indicating its ability to effectively utilize the knowledge learned from previous tasks. The contraction situation in Atlantis poses a huge challenge to CL agents, resulting in catastrophic forgetting of all methods. AACL achieves strong results in forward transfer metrics of both environments. While some methods excel in forgetting or forward transfer, they fail to balance both simultaneously. In contrast, AACL strikes a better balance between plasticity and stability. The continual return, a crucial metric for CL agents, varies significantly among different methods in these challenging experiments. Most methods exhibit low returns after all tasks, likely due to suffering from both catastrophic forgetting and plasticity loss (Abbas et al., 2023). Interestingly, FT, a naive method, performs better than other popular CL methods in Bigfish, highlighting the distinct challenges posed by CL-DC. AACL also achieves the best continual return in these experiments, demonstrating its robustness across different task complexities.

## 6 CONCLUSION

In this paper, we propose a new and practical problem called *Continual Learning with Dynamic Capabilities* (CL-DC), in which the agent's action space dynamically changes. This problem supplements the existing CL problem and provides more realistic situations for artificial intelligence systems. To tackle CL-DC, we introduce a new framework called **A**ction-**A**daptive **C**ontinual **L**earning (AACL) to enhance the generalization ability of CL agents across different action spaces. Inspired by the cortical functions that lead to consistent human behavior, this framework separates the agent's policy from the specific action spaces by building an action representation space. Additionally, we release a benchmark based on three environments to validate the effectiveness of popular CL methods in handling CL-DC. Extensive experimental results show the distinct challenge of CL-DC and demonstrate the superior performance of AACL compared to popular methods.

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

## A    APPENDIX: FRAMEWORK DETAILS

Algorithm 1 shows the complete process of AACL. The notations used in the algorithm are consistent with those in the main text. For each task, AACL consists of two stages: exploration and learning. The parameters $\phi$, $\delta$, and $\theta$ are continually updated in place as the process. After all tasks are completed, the policy $\pi_{E\theta}$ and decoder $g_\delta$ of the final task are returned. Therefore, we do not use superscript $i$ to denote the parameters in the algorithm.

---

**Algorithm 1** AACL

---

**Input:** Tasks with different action space $\{\mathcal{A}^i\}_{i=1}^N$.
**Initialize:** Policy parameter $\theta$, encoder parameter $\phi$ and decoder parameter $\delta$.
**for** $i = 1, 2, \ldots, N$ **do**
    See Task $i$ with action space $\mathcal{A}^i$

    **Exploration Stage**

    Use exploration policy to interact with the environment to collect transitions $\mathcal{T}^i$;
    **if** $i = 1$ **then**
        Update $\phi$ and $\delta$ with by minimizing Equation 3; // Encoder-decoder training
    **else**
        Update $\phi$ and $\delta$ with by minimizing Equation 6; // Action-adaptive fine-tuning

    **Learning Stage**

    Use Equation 4 to interact with the environment;
    Update $\theta$ by maximizing Equation 5; // Policy training

**Return:** Policy $\pi_{E\theta}$ and decoder $g_\delta$.

---

## B    APPENDIX: RELATED WORKS

### B.1    SELF-SUPERVISED LEARNING FOR REINFORCEMENT LEARNING

Existing reinforcement learning methods often require extensive data interactions with the environment, particularly in image-based RL tasks, which suffer from low sample efficiency and generalizability (Schrittwieser et al., 2020; Ye et al., 2020; Wang et al., 2024b). Recently, Self-Supervised Learning (SSL) has emerged to address these issues by learning a compact and informative representation of the environment (Li et al., 2022; Stooke et al., 2021). SSL approaches in RL encompass auxiliary tasks, contrastive learning, and data augmentation, each contributing to improved performance and efficiency.

Auxiliary tasks are a common approach in SSL for RL. These tasks include reconstruction loss, dynamics prediction, world modeling, and information-theoretic techniques to obtain efficient representations. Notable works in this area include GBRL, which uses a variational autoencoder to reconstruct state-action pairs (Liu et al., 2024), Dreamer, which is based on world models (Hafner et al., 2020), and Skew-Fit, which utilizes information-theoretic principles (Pong et al., 2020). These methods aim to enhance the agent's understanding of the environment by learning additional tasks that provide richer state representations.

Contrastive learning has gained traction in RL for its ability to learn valuable representations without requiring labeled data. Inspired by MoCo (He et al., 2020), CURL introduces an auxiliary contrastive learning task, enabling it to match the sample efficiency of state-based methods (Laskin et al., 2020a). Furthermore, it has been demonstrated that contrastive learning methods can be directly applied to action-labeled trajectories, achieving higher success rates in goal-conditioned RL tasks without additional data augmentation or auxiliary objectives (Eysenbach et al., 2022).

Data augmentation is another effective strategy in SSL for RL. DrQ exemplifies this approach by applying simple image augmentation techniques to standard model-free RL algorithms (Yarats et al., 2021). It enhances the robustness of RL from image inputs without requiring auxiliary losses or pretraining. Additionally, RAD explores general data augmentations for RL on both pixel-based and state-based inputs (Laskin et al., 2020b). To improve data efficiency and generalization (Laskin et al., 2020b). The introduction of new data augmentations significantly improves data efficiency and generalization.

While SSL for RL has significantly improved sample efficiency and generalization, the open research challenge of using SSL in CRL is an intriguing area that requires further exploration. Our proposed framework uses self-supervised learning to build an action representation space that decouples the agent's policy from the specific action space, enabling policy generalization.

## B.2 RL with Dynamic Action Spaces

Recent research has begun to address RL settings where the action space is not fixed but changes over time. Among these, LAICA (Chandak et al., 2020) and DAE (Ding et al., 2023) are particularly relevant to our work. However, there are fundamental differences that distinguish our approach from these prior efforts, both in terms of the problem formulation and the objectives.

LAICA addresses environments where the action space expands over time and introduces a probabilistic inverse dynamics model to facilitate adaptation to new actions. However, it focuses exclusively on action space expansion, without considering contraction or other types of changes, and does not address catastrophic forgetting. LAICA also prioritizes current action space performance, lacking mechanisms to preserve knowledge about previously available actions. In contrast, our framework tackles a more general continual learning problem, explicitly considering both expansion and contraction of the action space and maintaining performance across both current and historical action spaces. Technically, our encoder is a deterministic mapping, differing from LAICA's probabilistic approach, and we also introduce a dedicated benchmark to evaluate continual learning under diverse action space changes.

DAE explores incremental RL with expanding state and action spaces, aiming primarily to enhance exploration in new regions. Its methods are tailored to situations where the state and action spaces grow, but do not address contraction or the issue of catastrophic forgetting. Moreover, DAE's reliance on exploration strategies based on state-dependent probabilities is not applicable when the state space remains fixed and the action space contracts.

While LAICA and DAE have advanced the study of RL with dynamic action spaces, our work addresses a more general and realistic CL problem (CL-DC) which encompasses a wider variety of action space changes and explicitly tackles the challenge of catastrophic forgetting and forward transfer.

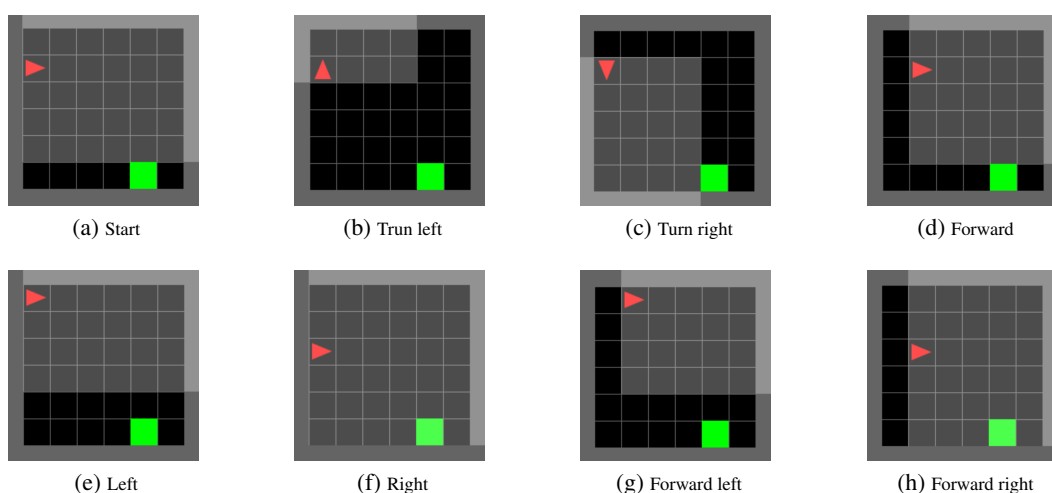

(a) Start  (b) Trun left  (c) Turn right  (d) Forward

(e) Left  (f) Right  (g) Forward left  (h) Forward right

Figure 7: The screenshots of actions in MiniGrid. The transparent white area represents the agent's field of view, which is the state. (a) The agent starts from this state. (b)–(h) The agent's state after the corresponding action.

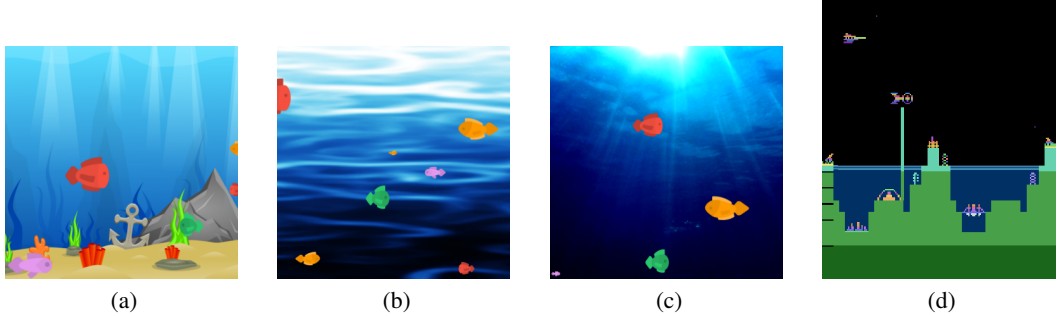

(a)  (b)  (c)  (d)

Figure 8: The screenshots of Bigfish and Atlantis. The texture and objects of Bigfish are procedurally generated.

## C  APPENDIX: ENVIRONMENTAL DETAILS

### C.1  ENVIRONMENTS AND TASK SEQUENCES.

In our experiments, we use three different environments to evaluate the performance of AACL: MiniGrid, Bigfish, and Atlantis. Although there are many environments in the Procgen (Cobbe et al., 2020) and Arcade Learning Environment (ALE) (Bellemare et al., 2013) benchmarks, they are not suitable for our problem setting. The reasons can be summarized as follows: 1) The change of action space in some environments can lead to non-smooth changes in agent's performance. For example, the contraction of action space in Backgammon and Breakout can cause the agent to be unable to complete the game. And the expansion of action space in DemonAttack and Galaxian (rightfire or leftfire) may not have an observable effect on the agent's performance. 2) Different actions have different effects on the agent's performance. In order to focus on studying the change of action space, we choose actions with similar properties for expansion and contraction.

**MiniGrid**[1] The MiniGrid contains a collection of simple 2D grid-world environments with a variety of objects, such as walls, doors, keys, and agents. These environments feature image-based partial observations, a discrete set of possible actions, and various objects characterized by their color and type. For expeditious training and evaluation, we only use the empty room environments of Mini-Grid in our experiments. By default, they have a discrete 7-dimensional action space and produce a 3-channel integer state encoding of the $7 \times 7$ grid directly including and in front of the agent.

---

[1]https://minigrid.farama.org/environments/minigrid/EmptyEnv/

Following the training setup for Atari (Schrittwieser et al., 2020), we modified the environments to output a $7 \times 7 \times 9$ by stacking three frames. Furthermore, we only use three basic movement actions from the original action space of MiniGrid: turn left, turn right, and move forward. Then, we expand the action space by adding four more actions to simulate CL-DC: move left, move right, forward left, forward right. The screenshots of these actions are shown in Figure 7. Finally, we design three tasks with different action spaces: a three-action task (turn left, turn right, forward), a five-action task (turn left, turn right, forward, left, right), and a seven-action task (turn left, turn right, forward, left, right, forward left, forward right).

**Bigfish.**[2] The Bigfish environment is part of the Procgen benchmark, which is a collection of procedurally generated environments designed to evaluate generalization in RL algorithms. Procgen was proposed as a replacement for the Atari games benchmark while being computationally faster to simulate than Atari. For faster training and evaluation, we chose Bigfish with the easiest configuration as the base environment in our experiments (set 'distribution_mode' to 'easy'), in which the agent starts as a small fish and needs to become bigger by eating other fish. We also set 'start_level' and 'num_levels' to 0, which means that each reset generates a new level based on a different random seed, making the agent's experience independent of any particular level. Figures 8a, 8b and 8c show the screenshots of this environment. The input observations are RGB images of dimension $64 \times 64 \times 3$, along with 15 possible discrete actions. Similar to MiniGrid, we only use nine basic movement actions from the original action space of Bigfish: stay, up, down, left, right, up-left, up-right, down-left, and down-right. Then, we design three tasks with different action spaces: a three-action task (stay, up, down), a five-action task (stay, up, down, left, right), and a nine-action task (stay, up, down, left, right, up-left, up-right, down-left, down-right).

**Atlantis.**[3] Atlantis is a classic Atari game provided by ALE (Bellemare et al., 2013). In this game, the agent controls three cannon at the bottom of the screen and must defend the city of Atlantis from alien invaders. The rewards are given based on the number of invaders destroyed, and the game ends when all seven installations are destroyed. As illustrated in Figure 8d, the input observations are RGB images of dimension $210 \times 160 \times 3$, and the action space consists of 4 discrete actions. By restricting the action space of the agent, we design a three-action task for Atlantis: a two-action task (noop, fire), a three-action task (noop, fire, rightfire), and a four-action task (noop, fire, rightfire, leftfire). Note that the fire actions (fire, rightfire, and leftfire) appear semantically similar, but their availability may significantly impact the agent's overall performance. This is because these actions are crucial for the agent's objective of preventing the destruction of the installations. For instance, if the agent is unable to execute rightfire, the installations on the left will be vulnerable to attacks from invaders approaching from that direction.

## C.2 COMPARED METHODS.

We compare AACL with nine methods in CL-DC. These methods cover a broad spectrum of CL strategies, including replay-based (both buffer and generative), regularization-based, architecture-based, and optimization-based approaches. The details of the methods are as follows:

- **IND.** This method represents a traditional DRL setup where an agent is trained independently on each task. This serves as a foundational comparison point to underscore the advantages of CL, as it lacks any mechanism for knowledge retention or transfer.

- **FT.** Building upon the standard DRL algorithm, this method differs from IND by using a single agent that is sequentially fine-tuned across different tasks. As a naive CL method, this method provides a basic measure of an agent's capacity to maintain knowledge of earlier tasks while encountering new tasks (Gaya et al., 2022).

- **CLEAR.** A classical CL method aiming to mitigate catastrophic forgetting by using a replay buffer to store experiences from previous tasks (Rolnick et al., 2019). It uses off-policy learning and behavioral cloning from replay to enhance stability, as well as on-policy learning to preserve plasticity.

---

[2]https://github.com/openai/procgen
[3]https://ale.farama.org/environments/atlantis/

Table 3: Network structure for MiniGrid. All convolutional layers use a kernel size of $2 \times 2$ and a stride of 1. Linear 2 and Linear 4 are the output heads in AACL, while Linear 3 and Linear 5 are the output heads in other methods.

|  | Layer | Input channels/units | Output channels/units |
|---|---|---|---|
| Backbone | Conv 1 | 9 | 32 |
|  | Conv 2 | 32 | 64 |
|  | Conv 3 | 64 | 128 |
|  | Linear 1 | 2048 | 64 |
| Value output | Linear 2 | 64+256+1 | 1 |
|  | Linear 3 | 64+7+1 | 1 |
| Policy output | Linear 4 | 64+256+1 | 256 |
|  | Linear 5 | 64+7+1 | 7 |
| Encoder | Conv 4 | 9 | 32 |
|  | Conv 5 | 32 | 64 |
|  | Conv 6 | 64 | 128 |
|  | Linear 6 | 2048 | 64 |
|  | Linear 7 | 64 | 256 |
| Decoder | Linear 8 | 256 | 7 |

- **EWC.** An RL implementation of Elastic Weight Consolidation (EWC) (Kirkpatrick et al., 2017), which is designed to mitigate catastrophic forgetting by selectively constraining the update of weights that are important for previous tasks.

- **Online-EWC.** A modified version of EWC that adds an explicit forgetting mechanism to perform well with low computational cost (Schwarz et al., 2018).

- **Mask.** A CL method that adapts modulating masks to the network architecture to prevent catastrophic forgetting (Ben-Iwhiwhu et al., 2023)[1]. The linear combination of the previously learned masks is used to exploit knowledge when learning new tasks.

- **P&C.** The Progress and Compress algorithm (Schwarz et al., 2018), which adopts a dual-architecture approach. It maintains a "knowledge base" to preserve learned skills and an "active column" for learning new tasks. Online EWC is part of this method.

- **UPGD.** A CL method designed to address the loss of plasticity and catastrophic forgetting. UPGD employs utility-based perturbed gradient descent to rejuvenate the network's learning capability, simultaneously mitigating catastrophic forgetting by maintaining the utility of critical connections (Elsayed & Mahmood, 2024). [2]

- **TDGR.** A generative replay method that addresses the storage constraints of buffer-based methods like CLEAR. Instead of storing raw experience, TDGR trains a generative model to synthesize trajectories of past tasks (Yue et al., 2025). [3]

Among the these methods, CLEAR is specifically designed for and implemented with IMPALA (Espeholt et al., 2018), and MASK also adopts IMPALA in its experiments. Other methods do not impose restrictions on the base RL algorithm. Therefore, we implemented all methods using IMPALA to ensure a fair comparison. Additionally, our benchmark is constructed based on the CORA platform (Powers et al., 2022), which utilizes IMPALA to accelerate training and facilitate large-scale experiments. This ensures consistency and comparability with established CRL benchmarks.

### C.3  Network Structures

All methods in our experiments are implemented based on IMPALA (Espeholt et al., 2018). The network of this algorithm is consistent across all methods, except for the specific components of each method. For MiniGrid, we use a small network as the input observation is an image with shape $9 \times 7 \times 7$. As shown in Table 3, each network consists of a convolutional neural network (CNN)

---

[1] We use the code at `https://github.com/dlpbc/mask-lrl-procgen/tree/develop_v2`

[2] We implemented UPGD based on the code at `https://github.com/mohmdelsayed/upgd`

[3] We implemented TDGR based on the code at `https://github.com/WilliamYue37/t-DGR`

Table 4: Network structure for Bigfish and Atlantis. All convolutional layers in the backbone use a kernel size of $3 \times 3$ and a stride of 1. The kernel sizes of the convolutional layers in the encoder are $8 \times 8$, $4 \times 4$, and $3 \times 3$, respectively. The stride of them are 4, 2, and 1, respectively. All maxpool layers use a kernel size of $3 \times 3$ and a stride of 2. Linear 2 and Linear 4 are the output heads in AACL, while Linear 3 and Linear 5 are the output heads in other methods.

| | Layer | Input channels/units | Output channels/units |
|---|---|---|---|
| | Conv 1 | 3 | 32 |
| | MaxPool | 32 | 32 |
| | Residual 1 | 32 | 32 |
| | Residual 2 | 32 | 32 |
| Backbone | Conv 2 | 32 | 64 |
| | MaxPool | 64 | 64 |
| | Residual 3 | 64 | 64 |
| | Reedisidual 4 | 64 | 64 |
| | Conv 3 | 64 | 64 |
| | MaxPool | 64 | 64 |
| | Residual 5 | 64 | 64 |
| | Reedisidual 6 | 64 | 64 |
| | Linear 1 | 3136 | 512 |
| Value | Linear 2 | 512+256+1 | 1 |
| output | Linear 3 | 512+9+1 | 1 |
| Policy | Linear 4 | 512+256+1 | 256 |
| output | Linear 5 | 512+9+1 | 7 |
| | Conv 4 | 9 | 32 |
| | Conv 5 | 32 | 64 |
| Encoder | Conv 6 | 64 | 64 |
| | Linear 6 | 1024 | 512 |
| | Linear 7 | 512 | 256 |
| Decoder | Linear 8 | 256 | 7 |

with three convolutional layers and two fully connected layers. ReLU activation is employed in all networks except the output layers of the policy network in AACL, which uses a sigmoid activation. Note that the number of input units for the policy and value output heads changes because the one-hot action vector and reward scalar from the previous time step are concatenated to the output of Linear 1. For Bigfish and Atlantis, we replace the CNN with the IMPALA architecture to improve the representation ability of bigger images. As shown in Table 4, this architecture consists of three IMPALA blocks, each of which contains a convolutional layer and two residual blocks. Additionally, we employ a bigger CNN in the encoder of AACL to extract features from the input image.

### C.4 HYPERPARAMETERS

The hyperparameters for the competitive experiments are presented in Table 5 and Table 6. Most values follow the settings in CORA (Powers et al., 2022). For TDGR and UPGD, we use the hyperparameters recommended in their official implementations. Note that for AACL, we did not conduct experiments to search for the best hyperparameters. Due to the small action spaces, we only use a random policy in the exploration stage of AACL. The number of exploration steps for AACL on new tasks is set to $10^4$. This parameter is relatively small compared to the number of training steps per task and is not being tuned. In the implementation of CLEAR, each actor only gets sampled once during training, so we need the same number of actors as well as batch size.

### C.5 METRICS

Based on the agent's normalized expected return, we evaluated the continual learning performance of our framework and other methods using the following metrics: continual return, forgetting, and forward transfer. Let us consider a sequence with $N$ tasks, where $P_{i,j}$ represents the performance (evaluation return) of task $j$ after the agent has been trained on task $i$. We normalize $P_{i,j}$ to $p_{i,j} \in [0, 1]$ by the maximum performance within the run Powers et al. (2022). Then, the above metrics can be defined:

Table 5: Hyperparameters for the experiments on MiniGrid. $\lambda$ is the regularization coefficient.

| Method | Hyperparameter | Value |
|---|---|---|
| Common | Num. of actors | 6 |
| | Num. of learner | 2 |
| | Batch size | 256 |
| | Learning rate | $4 \times 10^{-4}$ |
| | Entropy | 0.01 |
| | Rollout length | 20 |
| | Optimizer | RMSProp |
| | Discount factor | 0.99 |
| | Gradient clip | 40 |
| | Num. of training steps per task | $3 \times 10^6$ |
| | Num. of evaluation episodes | 10 |
| | Evaluation interval | $10^5$ |
| CLEAR | Num. of actors | 12 |
| | Batch size | 12 |
| | Replay buffer size | $5 \times 10^6$ |
| EWC | $\lambda$ | $10^4$ |
| | Replay buffer size | $10^6$ |
| | Min. frames per task | $2 \times 10^5$ |
| Online-EWC | $\lambda$ | 175 |
| | Replay buffer size | $10^6$ |
| P&C | $\lambda$ | 3000 |
| | Replay buffer size | $10^5$ |
| | Num. of progress train steps | 3906 |
| UPGD | Weight decay | $10^{-4}$ |
| | Beata utility | 0.999 |
| | $\sigma$ | 0.01 |
| | $\beta_1$ | 0.9 |
| | $\beta_2$ | 0.999 |
| | $\epsilon$ | $10^{-5}$ |
| TDGR | Horizon | 16 |
| | Ratio | 0.9 |
| | Timesteps | 1000 |
| | Diffusion steps | $10^4$ |
| | Warmup steps | $5 \times 10^4$ |
| | Epochs of learner | 300 |
| AACL | $\lambda$ | $2 \times 10^4$ |
| | Action representation size | 256 |
| | Exploration steps | $10^4$ |

Table 6: Hyperparameters for the experiments on Bigfish and Atlantis. Other hyperparameters are the same as those on MiniGrid.

| Hyperparameter | Value |
|---|---|
| Num. of actors | 21 |
| Num. of learner | 2 |
| Batch size | 32 |
| Num. of training steps per task | $5 \times 10^6$ |
| Num. of evaluation episodes | 10 |
| Evaluation interval | $2.5 \times 10^5$ |

- **Continual return**: The continual return for task $i$ is defined as:

$$\mathbf{R}_i := \frac{1}{i} \sum_{j=1}^{i} P_{i,j}. \tag{7}$$

This metric provides an overall view of the agent's performance up to and including task $i$. We do not use normalized performance here to directly reflect the agent's actual return and different environments' difficulties. The final value, $\mathbf{R}$ is a single-value summary of the agent's overall performance after all tasks and is included in the result tables.

- **Forgetting**: The forgetting for task $i$ measures the decline in performance for that task after training has concluded. It is calculated by:

$$\mathbf{F}_i := \frac{1}{i-1} \sum_{j=1}^{i-1} (p_{i-1,j} - p_{i,j}) \tag{8}$$

When $\mathbf{F}_i > 0$, the agent has become worse at the past tasks after training on new task $i$, indicating forgetting has occurred. Conversely, when $\mathbf{F}_i < 0$, the agent has become better at past tasks, indicating backward transfer has been observed. The overall forgetting metric, $\mathbf{F} \in [-1, 1]$, is the average of $\mathbf{F}_i$ values for all tasks, providing insight into how much knowledge the agent retains over time. We report $\mathbf{F}$ in the results tables.

- **Forward transfer**: The forward transfer for task $i$ quantifies the positive impact that learning task $i$ has on the performance of subsequent tasks. It is computed as follows:

$$\mathbf{T}_i := \frac{1}{N-i} \sum_{j=i+1}^{N} (p_{i,j} - p_{i-1,j}) \tag{9}$$

When $\mathbf{T}_i > 0$, the agent has become better at later tasks after training on earlier task $i$, indicating forward transfer has occurred through zero-shot learning. When $\mathbf{T}_i < 0$, the agent has become worse at later tasks, indicating negative transfer has occurred. The overall forward transfer, $\mathbf{T} \in [-1, 1]$, is the mean of $\mathbf{T}_i$ values across all tasks, providing insight into the generalization ability of the agent. We report $\mathbf{T}$ in the results tables.

## C.6 COMPUTE RESOURCES

Our code is implemented with Python 3.9.17 and Torch 2.0.1+cu118. Each method on MiniGrid was trained using an AMD Ryzen 5 3600 CPU (6 cores) with 32GB RAM and an NVIDIA GeForce RTX 1060 GPU. The Bigfish experiments were conducted on an AMD Ryzen 9 7950X CPU (16 cores) with 512GB RAM and an NVIDIA GeForce RTX 4070Ti Super GPU. The computing devices of the Atlantis experiments are the same as Bigfish, and the runtime is similar, so we do not show it. As illustrated in Figure 9a, each run, consisting of three MiniGrid tasks, takes about 1 hour to complete. However, there is a notable difference in the runtime of the methods when applied to Bigfish tasks, as shown in Figure 9b. Specifically, the CLEAR and Mask take approximately twice and four times as long as the baselines, respectively. This increased runtime may attribute to the additional computation required to update the replay buffer and masks. Although the runtime of AACL is longer than that of the baselines, it remains acceptable for practical applications when compared to CLEAR and Mask. In summary, the total runtime is influenced by four factors: the

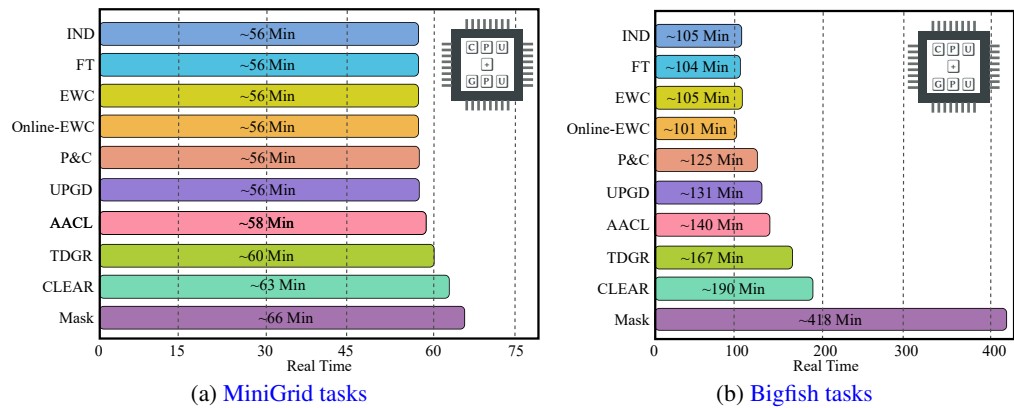

(a) MiniGrid tasks  (b) Bigfish tasks

Figure 9: Average runtime of seven methods on three MiniGrid tasks and three Bigfish tasks.

device, the domain, the computation time of the algorithm, and the behavior of the policy. Replay-based and architecture-based methods may experience a sharp increase in runtime due to heightened task complexity. In contrast, our method requires only a modest amount of additional computing resources to explore the environment, thereby achieving a balanced trade-off between efficiency and effectiveness.

# D  APPENDIX: ADDITIONAL EXPERIMENTS AND RESULTS

## D.1  DETAILED METRICS ON MINIGRID

Tables 7 and 8 present the detailed metrics of forgetting and forward transfer across three MiniGrid tasks in the situations of expansion and contraction. The columns represent trained tasks, while the rows represent evaluated tasks. We denote the task with an action space of size $n$ as "$n$-Actions". The average results across all tasks (bottom right of each subtable) are reported in the main text. In each forgetting table, negative values are shown in green and positive values in red, with darker shades representing larger magnitudes. Values close to zero are unshaded. In each forward transfer table, positive values are shown in green, and negative values in red.

These results further highlight the superiority of AACL. Additionally, they illustrate the difference between various action spaces and situations. Forgetting is slight in the expansion situation but more pronounced in the contraction situation. After training with 3-actions, the performance on previous tasks significantly degrades. However, forward transfer remains similar across different situations. In the expansion situation, training on 3-actions benefits evaluation performance on the subsequent tasks. In the contraction situation, training on 7-actions similarly benefits performance on subsequent tasks. This phenomenon may be attributed to the high similarity between tasks, where learning on the first task aids in learning subsequent tasks.

## D.2  DETAILED RESULTS ON MINIGRID

In Figure 4, the agent achieves nearly optimal performance on tasks 2 and 3 when trained on task 1. After adding new actions to the action space, the further improvement in performance is not significant. This raises the question of whether the agent is effectively utilizing the newly available actions to enhance its policy. Therefore, we provides more detailed results on the **expansion** situation in Table 9. In MiniGrid, the tasks are relatively simple, and the action spaces are small. Our results indicate that AACL is able to generalize the policy learned from task 1 to tasks 2 and 3 with high zero-shot performance. As shown in the left of Table 9, after training on Task 1, the agent already performs well on Tasks 2 and 3. However, there is still a measurable improvement when new actions are introduced and the agent trains on subsequent tasks. In more complex environments (Bigfish), the benefits of utilizing new actions are much more significant. After training on Task 2, the agent's performance on Task 2 increases by $10.641$ compared to Task 1. After training on Task 3, the performance further increases by $4.483$.

Table 7: Forgetting and forward transfer in the **expansion** situation across three MiniGrid tasks.

(a) IND

| | Forgetting | | | |
|---|---|---|---|---|
| | 3-Actions | 5-Actions | 7-Actions | Avg ± SEM |
| 3-Actions | – | $0.31 \pm 0.05$ | $-0.05 \pm 0.07$ | $0.13 \pm 0.02$ |
| 5-Actions | – | – | $-0.03 \pm 0.02$ | $-0.03 \pm 0.02$ |
| 7-Actions | – | – | – | – |
| Avg ± SEM | – | $0.31 \pm 0.05$ | $-0.04 \pm 0.04$ | $0.08 \pm 0.02$ |

(b) FT

| | Forgetting | | | | Forward Transfer | | | |
|---|---|---|---|---|---|---|---|---|
| | 3-Actions | 5-Actions | 7-Actions | Avg ± SEM | 3-Actions | 5-Actions | 7-Actions | Avg ± SEM |
| 3-Actions | – | $-0.03 \pm 0.02$ | $0.12 \pm 0.05$ | $0.05 \pm 0.03$ | – | – | – | – |
| 5-Actions | – | – | $0.01 \pm 0.02$ | $0.01 \pm 0.02$ | $0.69 \pm 0.05$ | – | – | $0.69 \pm 0.05$ |
| 7-Actions | – | – | – | – | $0.59 \pm 0.07$ | $0.15 \pm 0.09$ | – | $0.37 \pm 0.04$ |
| Avg ± SEM | – | $-0.03 \pm 0.02$ | $0.07 \pm 0.03$ | $0.03 \pm 0.02$ | $0.64 \pm 0.04$ | $0.15 \pm 0.09$ | – | $0.48 \pm 0.03$ |

(c) EWC

| | Forgetting | | | | Forward Transfer | | | |
|---|---|---|---|---|---|---|---|---|
| | 3-Actions | 5-Actions | 7-Actions | Avg ± SEM | 3-Actions | 5-Actions | 7-Actions | Avg ± SEM |
| 3-Actions | – | $-0.04 \pm 0.02$ | $-0.02 \pm 0.01$ | $-0.03 \pm 0.01$ | – | – | – | – |
| 5-Actions | – | – | $-0.06 \pm 0.03$ | $-0.06 \pm 0.03$ | $0.55 \pm 0.09$ | – | – | $0.55 \pm 0.09$ |
| 7-Actions | – | – | – | – | $0.64 \pm 0.06$ | $0.09 \pm 0.09$ | – | $0.36 \pm 0.04$ |
| Avg ± SEM | – | $-0.04 \pm 0.02$ | $-0.04 \pm 0.01$ | $-0.04 \pm 0.01$ | $0.60 \pm 0.07$ | $0.09 \pm 0.09$ | – | $0.43 \pm 0.05$ |

(d) Online-EWC

| | Forgetting | | | | Forward Transfer | | | |
|---|---|---|---|---|---|---|---|---|
| | 3-Actions | 5-Actions | 7-Actions | Avg ± SEM | 3-Actions | 5-Actions | 7-Actions | Avg ± SEM |
| 3-Actions | – | $-0.03 \pm 0.04$ | $0.04 \pm 0.03$ | $0.01 \pm 0.02$ | – | – | – | – |
| 5-Actions | – | – | $0.03 \pm 0.03$ | $0.03 \pm 0.03$ | $0.50 \pm 0.09$ | – | – | $0.50 \pm 0.09$ |
| 7-Actions | – | – | – | – | $0.58 \pm 0.09$ | $0.12 \pm 0.11$ | – | $0.35 \pm 0.05$ |
| Avg ± SEM | – | $-0.03 \pm 0.04$ | $0.04 \pm 0.02$ | $0.02 \pm 0.01$ | $0.54 \pm 0.08$ | $0.12 \pm 0.11$ | – | $0.40 \pm 0.05$ |

(e) CLEAR

| | Forgetting | | | | Forward Transfer | | | |
|---|---|---|---|---|---|---|---|---|
| | 3-Actions | 5-Actions | 7-Actions | Avg ± SEM | 3-Actions | 5-Actions | 7-Actions | Avg ± SEM |
| 3-Actions | – | $0.44 \pm 0.09$ | $0.03 \pm 0.13$ | $0.24 \pm 0.05$ | – | – | – | – |
| 5-Actions | – | – | $0.15 \pm 0.07$ | $0.15 \pm 0.07$ | $0.87 \pm 0.03$ | – | – | $0.87 \pm 0.03$ |
| 7-Actions | – | – | – | – | $0.87 \pm 0.02$ | $-0.01 \pm 0.01$ | – | $0.43 \pm 0.01$ |
| Avg ± SEM | – | $0.44 \pm 0.09$ | $0.09 \pm 0.09$ | $0.21 \pm 0.06$ | $0.87 \pm 0.02$ | $-0.01 \pm 0.01$ | – | $0.58 \pm 0.01$ |

(f) MASK

| | Forgetting | | | | Forward Transfer | | | |
|---|---|---|---|---|---|---|---|---|
| | 3-Actions | 5-Actions | 7-Actions | Avg ± SEM | 3-Actions | 5-Actions | 7-Actions | Avg ± SEM |
| 3-Actions | – | $0.01 \pm 0.11$ | $-0.15 \pm 0.07$ | $-0.07 \pm 0.05$ | – | – | – | – |
| 5-Actions | – | – | $0.02 \pm 0.07$ | $0.02 \pm 0.07$ | $0.03 \pm 0.03$ | – | – | $0.03 \pm 0.03$ |
| 7-Actions | – | – | – | – | $-0.04 \pm 0.05$ | $0.08 \pm 0.03$ | – | $0.02 \pm 0.03$ |
| Avg ± SEM | – | $0.01 \pm 0.11$ | $-0.06 \pm 0.06$ | $-0.04 \pm 0.04$ | $-0.00 \pm 0.03$ | $0.08 \pm 0.03$ | – | $0.02 \pm 0.03$ |

(g) AACL

| | Forgetting | | | | Forward Transfer | | | |
|---|---|---|---|---|---|---|---|---|
| | 3-Actions | 5-Actions | 7-Actions | Avg ± SEM | 3-Actions | 5-Actions | 7-Actions | Avg ± SEM |
| 3-Actions | – | $-0.01 \pm 0.02$ | $-0.01 \pm 0.02$ | $-0.01 \pm 0.01$ | – | – | – | – |
| 5-Actions | – | – | $-0.04 \pm 0.04$ | $-0.04 \pm 0.04$ | $0.88 \pm 0.03$ | – | – | $0.88 \pm 0.03$ |
| 7-Actions | – | – | – | – | $0.89 \pm 0.04$ | $-0.05 \pm 0.02$ | – | $0.42 \pm 0.02$ |
| Avg ± SEM | – | $-0.01 \pm 0.02$ | $-0.02 \pm 0.02$ | $-0.02 \pm 0.01$ | $0.88 \pm 0.03$ | $-0.05 \pm 0.02$ | – | $0.57 \pm 0.02$ |

Table 8: Forgetting and forward transfer in the **contraction** situation across three MiniGrid tasks.

(a) IND

|  | Forgetting | | | |
|---|---|---|---|---|
|  | 7-Actions | 5-Actions | 3-Actions | Avg ± SEM |
| 7-Actions | – | $0.06 \pm 0.02$ | $0.25 \pm 0.09$ | $0.15 \pm 0.04$ |
| 5-Actions | – | – | $0.48 \pm 0.08$ | $0.48 \pm 0.08$ |
| 3-Actions | – | – | – | – |
| Avg ± SEM | – | $0.06 \pm 0.02$ | $0.37 \pm 0.08$ | $0.26 \pm 0.05$ |

(b) FT

|  | Forgetting | | | | Forward Transfer | | | |
|---|---|---|---|---|---|---|---|---|
|  | 7-Actions | 5-Actions | 3-Actions | Avg ± SEM | 7-Actions | 5-Actions | 3-Actions | Avg ± SEM |
| 7-Actions | – | $0.01 \pm 0.01$ | $0.60 \pm 0.10$ | $0.31 \pm 0.05$ | – | – | – | – |
| 5-Actions | – | – | $0.56 \pm 0.10$ | $0.56 \pm 0.10$ | $0.82 \pm 0.04$ | – | – | $0.82 \pm 0.04$ |
| 3-Actions | – | – | – | – | $0.42 \pm 0.07$ | $-0.07 \pm 0.08$ | – | $0.18 \pm 0.03$ |
| Avg ± SEM | – | $0.01 \pm 0.01$ | $0.58 \pm 0.09$ | $0.39 \pm 0.06$ | $0.62 \pm 0.05$ | $-0.07 \pm 0.08$ | – | $0.39 \pm 0.03$ |

(c) EWC

|  | Forgetting | | | | Forward Transfer | | | |
|---|---|---|---|---|---|---|---|---|
|  | 7-Actions | 5-Actions | 3-Actions | Avg ± SEM | 7-Actions | 5-Actions | 3-Actions | Avg ± SEM |
| 7-Actions | – | $-0.01 \pm 0.02$ | $0.63 \pm 0.13$ | $0.31 \pm 0.07$ | – | – | – | – |
| 5-Actions | – | – | $0.59 \pm 0.12$ | $0.59 \pm 0.12$ | $0.88 \pm 0.03$ | – | – | $0.88 \pm 0.03$ |
| 3-Actions | – | – | – | – | $0.54 \pm 0.07$ | $-0.02 \pm 0.06$ | – | $0.26 \pm 0.03$ |
| Avg ± SEM | – | $-0.01 \pm 0.02$ | $0.61 \pm 0.12$ | $0.40 \pm 0.09$ | $0.71 \pm 0.04$ | $-0.02 \pm 0.06$ | – | $0.47 \pm 0.03$ |

(d) Online-EWC

|  | Forgetting | | | | Forward Transfer | | | |
|---|---|---|---|---|---|---|---|---|
|  | 7-Actions | 5-Actions | 3-Actions | Avg ± SEM | 7-Actions | 5-Actions | 3-Actions | Avg ± SEM |
| 7-Actions | – | $0.06 \pm 0.05$ | $0.45 \pm 0.10$ | $0.26 \pm 0.05$ | – | – | – | – |
| 5-Actions | – | – | $0.51 \pm 0.12$ | $0.51 \pm 0.12$ | $0.85 \pm 0.05$ | – | – | $0.85 \pm 0.05$ |
| 3-Actions | – | – | – | – | $0.47 \pm 0.03$ | $-0.00 \pm 0.06$ | – | $0.23 \pm 0.03$ |
| Avg ± SEM | – | $0.06 \pm 0.05$ | $0.48 \pm 0.10$ | $0.34 \pm 0.06$ | $0.66 \pm 0.04$ | $-0.00 \pm 0.06$ | – | $0.44 \pm 0.03$ |

(e) CLEAR

|  | Forgetting | | | | Forward Transfer | | | |
|---|---|---|---|---|---|---|---|---|
|  | 7-Actions | 5-Actions | 3-Actions | Avg ± SEM | 7-Actions | 5-Actions | 3-Actions | Avg ± SEM |
| 7-Actions | – | $0.29 \pm 0.14$ | $0.66 \pm 0.15$ | $0.47 \pm 0.01$ | – | – | – | – |
| 5-Actions | – | – | $0.80 \pm 0.07$ | $0.80 \pm 0.07$ | $0.85 \pm 0.03$ | – | – | $0.85 \pm 0.03$ |
| 3-Actions | – | – | – | – | $0.64 \pm 0.05$ | $-0.12 \pm 0.08$ | – | $0.26 \pm 0.03$ |
| Avg ± SEM | – | $0.29 \pm 0.14$ | $0.73 \pm 0.11$ | $0.58 \pm 0.03$ | $0.75 \pm 0.03$ | $-0.12 \pm 0.08$ | – | $0.46 \pm 0.02$ |

(f) MASK

|  | Forgetting | | | | Forward Transfer | | | |
|---|---|---|---|---|---|---|---|---|
|  | 7-Actions | 5-Actions | 3-Actions | Avg ± SEM | 7-Actions | 5-Actions | 3-Actions | Avg ± SEM |
| 7-Actions | – | $-0.08 \pm 0.12$ | $0.05 \pm 0.09$ | $-0.02 \pm 0.06$ | – | – | – | – |
| 5-Actions | – | – | $-0.03 \pm 0.07$ | $-0.03 \pm 0.07$ | $0.04 \pm 0.05$ | – | – | $0.04 \pm 0.05$ |
| 3-Actions | – | – | – | – | $0.10 \pm 0.05$ | $0.12 \pm 0.06$ | – | $0.11 \pm 0.03$ |
| Avg ± SEM | – | $-0.08 \pm 0.12$ | $0.01 \pm 0.07$ | $-0.02 \pm 0.04$ | $0.07 \pm 0.03$ | $0.12 \pm 0.06$ | – | $0.08 \pm 0.03$ |

(g) AACL

|  | Forgetting | | | | Forward Transfer | | | |
|---|---|---|---|---|---|---|---|---|
|  | 7-Actions | 5-Actions | 3-Actions | Avg ± SEM | 7-Actions | 5-Actions | 3-Actions | Avg ± SEM |
| 7-Actions | – | $-0.01 \pm 0.02$ | $-0.01 \pm 0.01$ | $-0.01 \pm 0.01$ | – | – | – | – |
| 5-Actions | – | – | $0.15 \pm 0.08$ | $0.15 \pm 0.08$ | $0.90 \pm 0.02$ | – | – | $0.90 \pm 0.02$ |
| 3-Actions | – | – | – | – | $0.91 \pm 0.02$ | $-0.01 \pm 0.02$ | – | $0.45 \pm 0.01$ |
| Avg ± SEM | – | $-0.01 \pm 0.02$ | $0.07 \pm 0.04$ | $0.04 \pm 0.03$ | $0.90 \pm 0.01$ | $-0.01 \pm 0.02$ | – | $0.60 \pm 0.01$ |

Table 9: Detailed continual returns of AACL on three MiniGrid or Bigfish tasks in the **expansion** situation. AACL is evaluated after continual training on each task.

| Train/Eval Task | MiniGrid | | | BigFish | | |
|---|---|---|---|---|---|---|
| | Task 1 | Task 2 | Task 3 | Task 1 | Task 2 | Task 3 |
| Task 1 | $0.860 \pm 0.014$ | $0.885 \pm 0.008$ | $0.905 \pm 0.005$ | $6.495 \pm 0.543$ | $7.444 \pm 1.841$ | $5.525 \pm 1.200$ |
| Task 2 | $0.904 \pm 0.053$ | $0.908 \pm 0.086$ | $0.891 \pm 0.020$ | $8.606 \pm 2.113$ | $17.136 \pm 1.340$ | $20.289 \pm 2.138$ |
| Task 3 | $0.886 \pm 0.019$ | $0.914 \pm 0.006$ | $0.923 \pm 0.006$ | $10.615 \pm 1.894$ | $18.249 \pm 2.036$ | $21.619 \pm 1.094$ |

### D.3 COMBINED SITUATIONS

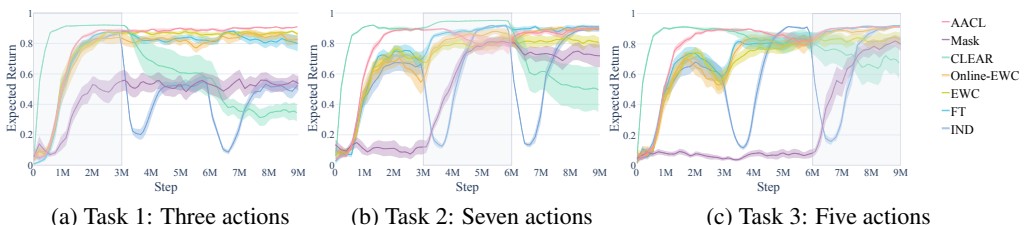

(a) Task 1: Three actions     (b) Task 2: Seven actions     (c) Task 3: Five actions

Figure 10: Performance of seven methods on three **MiniGrid** tasks in the **expansion & contraction** situation.

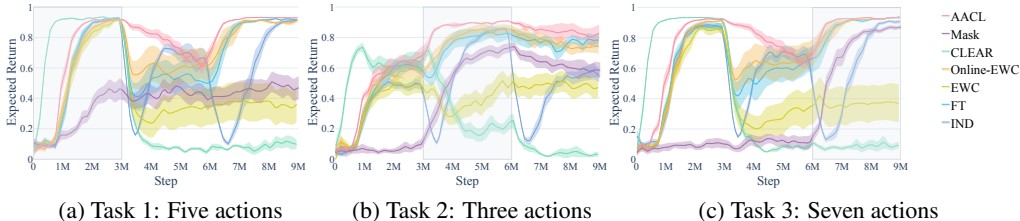

(a) Task 1: Five actions     (b) Task 2: Three actions     (c) Task 3: Seven actions

Figure 11: Performance of seven methods on three **MiniGrid** tasks in the **contraction & expansion** situation.

Figures 10 and 11, along with Table 10, show the performance of CL-DC and other CL methods in combined situations of expansion and contraction across three MiniGrid tasks. We use "expansion & contraction" to represent the situation where the size of action space expands to seven after the first task and contracts to five after the second task. Similarly, "contraction & expansion" represents the situation where the size of action space contracts to three after the first task and expands to seven after the second task. AACL achieves near-best performance in terms of continual return, forgetting, and forward transfer in both situations. These results are consistent with the previous experiments, further demonstrating the advantages of AACL in handling more complex situations.

### D.4 SCALING TO LONGER SEQUENCE

We also evaluate the CL methods' performance over a longer sequence of CL-DC. This sequence consists of five MiniGrid tasks, where the action space is expanding and then contracting. Each method is trained for a total of 15M environment steps, and results are reported over five runs. As shown in Figure 12 and Table 11, most methods' performance remains consistent with previous experiments, except for EWC, which performs better in this longer sequence. This improvement may be due to the repeated tasks in the sequence, benefiting EWC's regularization. AACL achieves the best performance in terms of continual return, forgetting, and forward transfer, demonstrating its ability to handle combined situations of action space changes over longer task sequences.

To further examine scalability, we extend the longer sequence experiment by repeating the five-task sequence four times, yielding a sequence of length 20. We compare AACL with online-EWC. Each method is trained for 60M environment steps, and all results are averaged over three runs. As shown in Table12, AACL maintains superior performance in terms of continual return and forward transfer, with comparable (and small) forgetting. These preliminary results indicate that AACL can scales to longer sequences while preserving stability and plasticity.

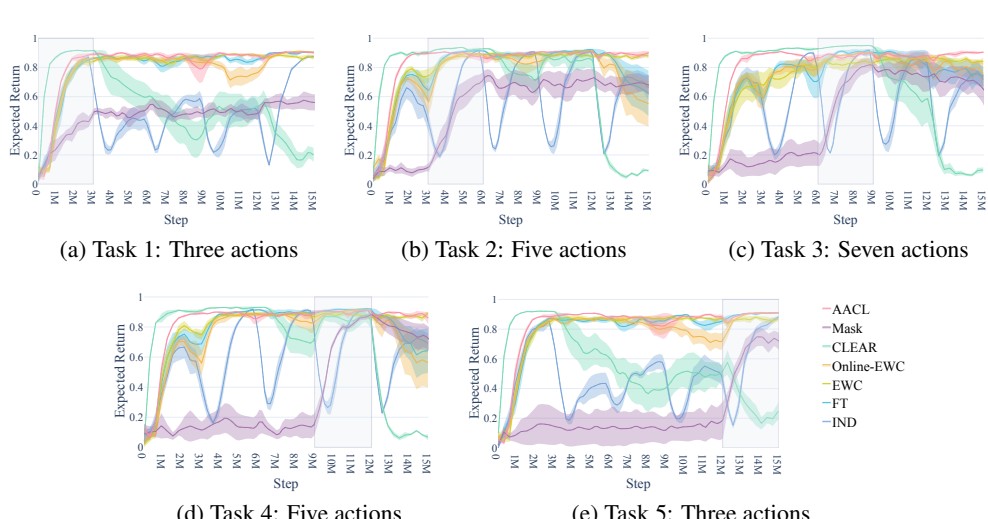

(a) Task 1: Three actions  (b) Task 2: Five actions  (c) Task 3: Seven actions

(d) Task 4: Five actions  (e) Task 5: Three actions

Figure 12: Performance of seven methods in a longer sequence (five MiniGrid tasks).

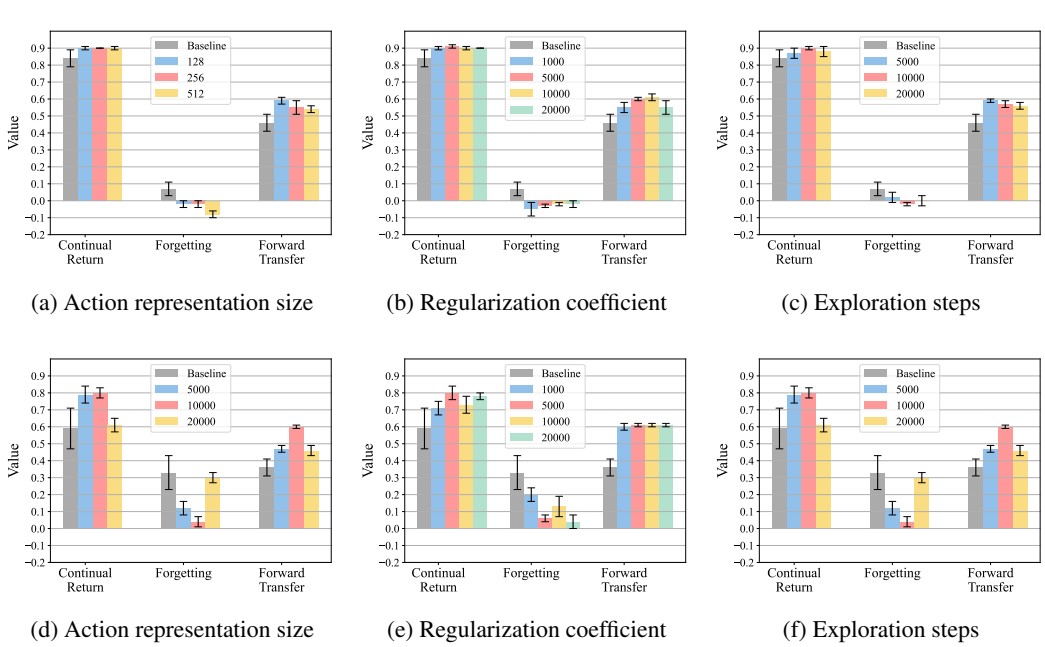

(a) Action representation size  (b) Regularization coefficient  (c) Exploration steps

(d) Action representation size  (e) Regularization coefficient  (f) Exploration steps

Figure 13: Hyperparameter sensitivity analysis for AACL across three MiniGrid tasks in the situations of expansion (**above**) and contraction (**bottom**). We examine the impact of the action representation size, the regularization coefficient ($\lambda$), and the exploration steps in AACL. Other hyperparameters are kept consistent with the competitive experiments, except that each experiment is repeated only five times. Error bars represent the standard error.

Table 10: Continual learning metrics of seven methods and two variants of AACL across three **Min-iGrid** tasks in combined situations of **expansion and contraction**. Continual return and forward transfer are abbreviated as "Return" and "Transfer", respectively. The top three results are highlighted in green, and the depth of the color indicates the ranking.

| Methods | Expansion & Contraction | | | Contraction & Expansion | | |
|---|---|---|---|---|---|---|
| | Return↑ | Forgetting↓ | Transfer↑ | Return↑ | Forgetting↓ | Transfer↑ |
| IND | $0.78 \pm 0.03$ | $0.12 \pm 0.02$ | – | $0.79 \pm 0.03$ | $0.10 \pm 0.02$ | – |
| FT | $0.87 \pm 0.03$ | $0.04 \pm 0.02$ | $0.50 \pm 0.02$ | $0.88 \pm 0.03$ | $0.02 \pm 0.01$ | $0.39 \pm 0.05$ |
| EWC | $0.83 \pm 0.03$ | $0.01 \pm 0.02$ | $0.44 \pm 0.04$ | $0.40 \pm 0.11$ | $0.20 \pm 0.05$ | $0.30 \pm 0.06$ |
| online-EWC | $0.88 \pm 0.02$ | $-0.02 \pm 0.02$ | $0.45 \pm 0.04$ | $0.86 \pm 0.03$ | $0.04 \pm 0.02$ | $0.40 \pm 0.05$ |
| Mask | $0.68 \pm 0.06$ | $0.04 \pm 0.04$ | $0.01 \pm 0.02$ | $0.64 \pm 0.05$ | $0.03 \pm 0.02$ | $0.05 \pm 0.03$ |
| CLEAR | $0.51 \pm 0.11$ | $0.35 \pm 0.06$ | $0.54 \pm 0.02$ | $0.07 \pm 0.02$ | $0.39 \pm 0.02$ | $0.21 \pm 0.03$ |
| AACL | $0.91 \pm 0.01$ | $-0.03 \pm 0.01$ | $0.55 \pm 0.02$ | $0.90 \pm 0.03$ | $0.03 \pm 0.02$ | $0.42 \pm 0.03$ |

Table 11: Continual learning metrics of seven methods in a longer sequence (five MiniGrid tasks). The top three results are highlighted in green, and the depth of the color indicates the ranking.

| Methods | Metrics | | |
|---|---|---|---|
| | Return↑ | Forgetting↓ | Transfer↑ |
| IND | $0.77 \pm 0.06$ | $0.08 \pm 0.02$ | – |
| FT | $0.76 \pm 0.10$ | $0.09 \pm 0.04$ | $0.29 \pm 0.01$ |
| EWC | $0.87 \pm 0.02$ | $0.00 \pm 0.00$ | $0.32 \pm 0.01$ |
| online-EWC | $0.74 \pm 0.11$ | $0.08 \pm 0.05$ | $0.16 \pm 0.01$ |
| Mask | $0.67 \pm 0.07$ | $0.05 \pm 0.03$ | $0.04 \pm 0.02$ |
| CLEAR | $0.14 \pm 0.04$ | $0.34 \pm 0.01$ | $0.29 \pm 0.02$ |
| AACL | $0.89 \pm 0.02$ | $-0.01 \pm 0.01$ | $0.34 \pm 0.01$ |

## D.5 HYPERPARAMETER SENSITIVITY ABLATION ANALYSIS

Figure 13 presents a hyperparameter sensitivity analysis of the action representation size, the regularization coefficient ($\lambda$), and the exploration steps in AACL across three MiniGrid tasks. For comparison, the performance of FT (baseline) is also shown. In the expansion situation, both hyperparameters have minimal impact on the performance of AACL. In the contraction situation, both hyperparameters slightly affect results. Forward transfer is positively correlated with the action representation size, but a smaller action representation size (128) may lead to a better continual return. A very small regularization coefficient (1000) can cause catastrophic forgetting and decreased continual return, further indicating the importance of regularization in the contraction situation.

For exploration steps, Our results indicate that the SSL process for building the action representation space is not overly demanding. Even with only $5 \times 10^3$ exploration steps, the agent can acquire a sufficiently rich representation for policy learning. In the expansion, reducing the exploration steps by half does not lead to a significant drop in continual return. In fact, it slightly increases forward transfer, though continual return is marginally lower. In the contraction, a shorter exploration phase slightly increases forgetting. Interestingly, increasing the number of exploration steps to $2 \times 10^4$ leads to a noticeable reduction in performance in the contraction situation. The forgetting increases substantially, and forward transfer drops sharply. We hypothesize that excessive exploration steps may cause the encoder-decoder to overfit to the current action space, thereby reducing the generalizability of the learned action representations across different action spaces. This is particularly problematic in the contraction setting, where the action space shrinks, and overfitting to a larger set may hinder adaptation. Overall, the contraction situation is more sensitive to hyperparameters than the expansion situation, but AACL's performance remains relatively robust in both.

## D.6 VISUALIZATION

To observe how action representation space learned by AACL changes with action spaces, we adopt t-SNE (Maaten & Hinton, 2008) to visualize the learned action representations on MiniGrid tasks in a 2D plane. Figure 14 shows that AACL constructs a smooth representation space, where points with similar influence are clustered together. For instance, the points of the action "turn left" and "turn right" are close to each other. Although very similar actions are not well distinguished in the figure, this reflects their substitutability. Through regularized constraints, as the action space

Table 12: Continual learning metrics on a longer sequence (five MiniGrid tasks repeated four times, sequence length 20).

| Methods | Metrics | | |
|---------|---------|---------|---------|
| | Return↑ | Forgetting↓ | Transfer↑ |
| online-EWC | $0.73 \pm 0.21$ | $0.03 \pm 0.02$ | $0.29 \pm 0.03$ |
| AACL | $0.80 \pm 0.19$ | $0.03 \pm 0.07$ | $0.35 \pm 0.00$ |

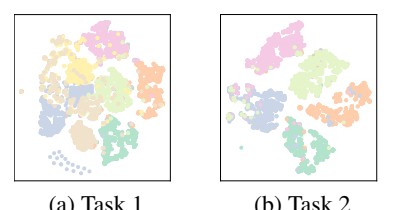 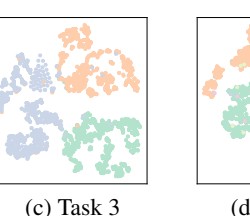 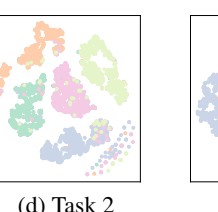 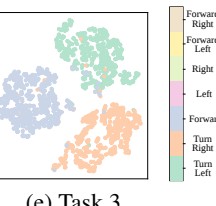

| (a) Task 1 | (b) Task 2 | (c) Task 3 | (d) Task 2 | (e) Task 3 |

Figure 14: 2D t-SNE visualizations of learned action representations on MiniGrid tasks, colored by actual actions. The number of points for each action is 1000. **(a)–(c)**: Fine-tuning with regularization. **(d)–(e)**: Learning independently.

changes, removed actions are replaced by existing actions while maintaining their relative positions in the previous action space. In contrast, independently learned representations are redistributed, failing to preserve previous action relationships. This indicates that action-adaptive fine-tuning in AACL can maintain knowledge of the previous action space, benefiting continual learning.

### D.7 EXPLORATION POLICY

To evaluate the impact of the exploration policy on the performance of AACL, we conduct comparisons between a random policy (AACL) and the policy from the previous task (AACL-L) on MiniGrid tasks. Other experiment settings remain the same with the main experiments. As shown in Table 13, the previous policy (AACL-L) performs worse than the random policy (AACL) in the expansion and contraction situations. This may be because the previous policy may introduce the absence of biases, which can adversely affect the representation learning.

Table 13: Continual learning metrics of two exploration policy of AACL in the **expansion** and the **contraction** situations on three **MiniGrid** tasks.

| Methods | Expansion & Contraction | | | Contraction & Expansion | | |
|---------|---------|---------|---------|---------|---------|---------|
| | Return↑ | Forgetting↓ | Transfer↑ | Return↑ | Forgetting↓ | Transfer↑ |
| AACL-L | $0.88 \pm 0.02$ | $0.01 \pm 0.01$ | $\mathbf{0.60 \pm 0.01}$ | $0.67 \pm 0.05$ | $0.19 \pm 0.04$ | $0.59 \pm 0.01$ |
| AACL | $\mathbf{0.90 \pm 0.01}$ | $\mathbf{-0.02 \pm 0.01}$ | $0.57 \pm 0.02$ | $\mathbf{0.80 \pm 0.03}$ | $\mathbf{0.04 \pm 0.03}$ | $\mathbf{0.60 \pm 0.01}$ |

### D.8 EXPANSION SITUATION ON BIGFISH

Figure 15 and Table 14 show the performance and metrics of AACL and other methods in the expansion situation across three Bigfish tasks. AACL achieves the best performance in terms of continual return and forward transfer. In this situation, all methods perform well in terms of mitigating forgetting, which is consistent with the results on MiniGrid tasks. Similarly, the overall performance of CLEAR is better than other methods, but AACL still outperforms it in CL metrics. In addition, the performance of all methods is improved with the expansion of the action space, as the agent has more actions to choose from.

### D.9 EXPERIMENTS ON ATLANTIS

Figure 16 show the performance of AACL and other methods in the contraction situation across three Atlantis tasks. As discussed in the main text, AACL demonstrates superior performance and reduced fluctuations in comparison to other methods. Furthermore, the timing of performance degradation among these methods during these tasks differs from that observed in Bigfish. This variability highlights the challenges inherent in generalizing policies across diverse actions. Accordingly, the

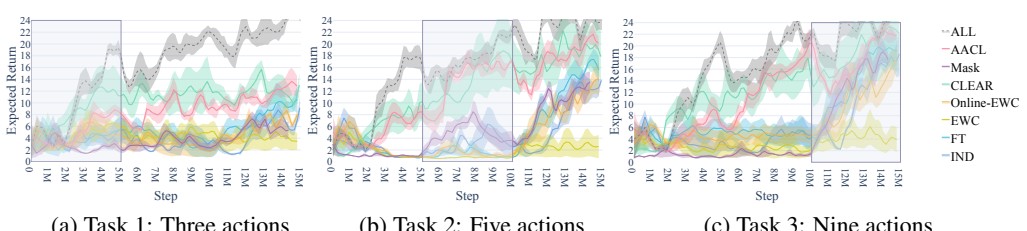

(a) Task 1: Three actions     (b) Task 2: Five actions     (c) Task 3: Nine actions

Figure 15: Performance of eight methods on three **Bigfish** tasks in the **expansion** situation.

Table 14: Continual learning metrics of eight methods in the **expansion** situation on three **Bigfish** tasks. The average continual return of ALL is 24.77, which is not provided in the table. The top three results are highlighted in green, and the depth of the color indicates the ranking.

| Methods | Metrics | | |
|---|---|---|---|
| | Return↑ | Forgetting↓ | Transfer↑ |
| IND | $13.86 \pm 2.41$ | $-0.27 \pm 0.08$ | – |
| FT | $15.13 \pm 0.10$ | $-0.25 \pm 0.04$ | $0.00 \pm 0.02$ |
| EWC | $3.31 \pm 1.88$ | $-0.00 \pm 0.11$ | $-0.08 \pm 0.12$ |
| online-EWC | $13.43 \pm 2.02$ | $-0.31 \pm 0.09$ | $0.02 \pm 0.05$ |
| Mask | $12.18 \pm 1.45$ | $-0.19 \pm 0.04$ | $-0.02 \pm 0.03$ |
| CLEAR | $17.68 \pm 4.79$ | $-0.04 \pm 0.03$ | $0.18 \pm 0.07$ |
| AACL | $18.27 \pm 1.22$ | $-0.05 \pm 0.11$ | $0.21 \pm 0.05$ |

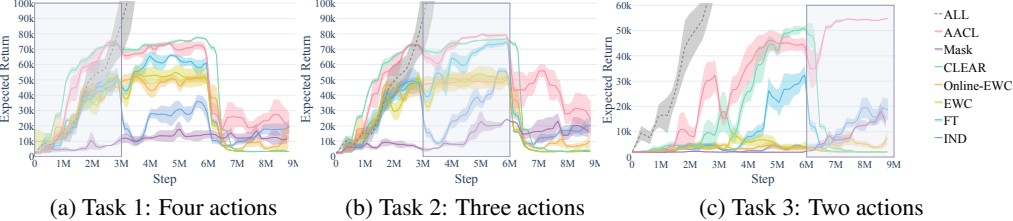

(a) Task 1: Four actions     (b) Task 2: Three actions     (c) Task 3: Two actions

Figure 16: Performance of eight methods on three **Atlantis** tasks in the **contraction** situation.

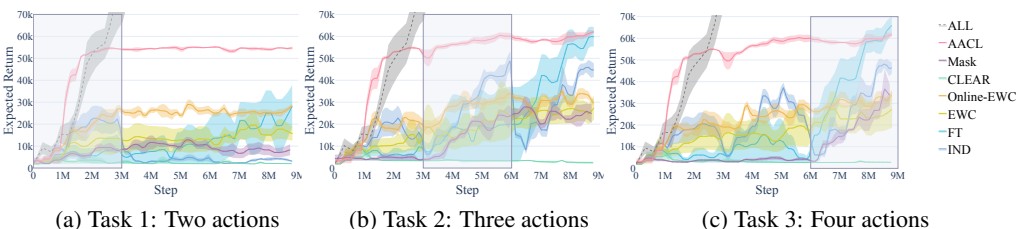

(a) Task 1: Two actions     (b) Task 2: Three actions     (c) Task 3: Four actions

Figure 17: Performance of eight methods on three **Atlantis** tasks in the **expansion** situation.

influence of the interplay among various actions on policy generalization will serve as the focus of our forthcoming research efforts.

Figure 17 and Table 15 present the performance and corresponding metrics of AACL alongside other methods in the expansion situation across three Atlantis tasks. Notably, the expected return curves for most methods in these tasks exhibit significant deviations compared to those observed in Bigfish, underscoring the unique challenges posed by this environment and highlighting the rapid learning capabilities of our framework. While AACL demonstrates some degree of plasticity loss and catastrophic forgetting relative to the FT, potentially attributable to insufficient training or overly stringent regularization, it still achieves state-of-the-art performance in terms of continual return and forward transfer.

Table 15: Continual learning metrics of eight methods in the **expansion** situation on three **Atlantis** tasks. The average continual return of ALL is $296, 949$, which is not provided in the table. The top three results are highlighted in green, and the depth of the color indicates the ranking.

| Methods | Metrics | | |
|---|---|---|---|
| | Return↑ | Forgetting↓ | Transfer↑ |
| IND | $31396 \pm 2339$ | $0.19 \pm 0.04$ | – |
| FT | $51312 \pm 6562$ | $-0.29 \pm 0.05$ | $0.05 \pm 0.04$ |
| EWC | $23449 \pm 6125$ | $-0.05 \pm 0.06$ | $0.17 \pm 0.08$ |
| online-EWC | $30593 \pm 1824$ | $0.04 \pm 0.05$ | $0.31 \pm 0.01$ |
| Mask | $22310 \pm 4109$ | $-0.04 \pm 0.10$ | $-0.00 \pm 0.02$ |
| CLEAR | $2425 \pm 270$ | $0.04 \pm 0.03$ | $0.05 \pm 0.00$ |
| AACL | $59498 \pm 656$ | $-0.01 \pm 0.00$ | $0.54 \pm 0.01$ |

## D.10 ADDITIONAL COMPARATIVE EXPERIMENTS ON MINIGRID

Table 16 summarizes the continual learning metrics on MiniGrid for both expansion and contraction settings, incorporating the three additional baselines: TDGR(Yue et al., 2025), UPGD(Elsayed & Mahmood, 2024), and P&C(Schwarz et al., 2018). Consistent with the main text, AACL attains the highest forward transfer in both settings, while maintaining strong average continual return and competitive forward transfer scores. Figures 18 19 show the performance curves of all methods, including the additional baselines, on MiniGrid tasks in expansion and contraction scenarios, respectively. Table 17 presents the corresponding metrics for Bigfish tasks, where AACL continues to demonstrate superior performance across all continual return in both expansion and contraction scenarios. Figures 20 and 21 illustrate the performance curves of all methods on Bigfish tasks in expansion and contraction scenarios, respectively.

Table 16: Continual learning metrics of ten methods in the **expansion** and **contraction** situations on three **MiniGrid** tasks. The added baselines include TDGR, UPGD, and P&C. Results are reported as mean $\pm$ standard deviation over ten seeds. The average continual return of ALL is $0.94$, which is not provided in the table. The top three results of compared methods are highlighted in green, and the depth of the color indicates the ranking.

| Methods | Expansion | | | Contraction | | |
|---|---|---|---|---|---|---|
| | Return↑ | Forgetting↓ | Transfer↑ | Return↑ | Forgetting↓ | Transfer↑ |
| IND | $0.81 \pm 0.02$ | $0.08 \pm 0.02$ | – | $0.67 \pm 0.06$ | $0.26 \pm 0.05$ | – |
| FT | $0.86 \pm 0.03$ | $0.03 \pm 0.02$ | $0.48 \pm 0.03$ | $0.52 \pm 0.07$ | $0.39 \pm 0.06$ | $0.39 \pm 0.03$ |
| EWC | $0.81 \pm 0.04$ | $-0.04 \pm 0.01$ | $0.43 \pm 0.05$ | $0.39 \pm 0.11$ | $0.40 \pm 0.09$ | $0.47 \pm 0.03$ |
| online-EWC | $0.87 \pm 0.02$ | $0.02 \pm 0.01$ | $0.40 \pm 0.05$ | $0.56 \pm 0.09$ | $0.34 \pm 0.06$ | $0.44 \pm 0.03$ |
| Mask | $0.72 \pm 0.05$ | $-0.04 \pm 0.04$ | $0.02 \pm 0.03$ | $0.70 \pm 0.06$ | $-0.02 \pm 0.04$ | $0.08 \pm 0.03$ |
| CLEAR | $0.73 \pm 0.06$ | $0.21 \pm 0.06$ | $0.58 \pm 0.01$ | $0.11 \pm 0.02$ | $0.58 \pm 0.03$ | $0.46 \pm 0.02$ |
| P&C | $0.59 \pm 0.08$ | $0.17 \pm 0.08$ | $0.36 \pm 0.05$ | $0.69 \pm 0.07$ | $0.10 \pm 0.05$ | $0.40 \pm 0.07$ |
| TDGR | $0.66 \pm 0.03$ | $0.33 \pm 0.03$ | $0.30 \pm 0.02$ | $0.74 \pm 0.17$ | $0.18 \pm 0.06$ | $0.47 \pm 0.04$ |
| UPGD | $0.73 \pm 0.08$ | $0.17 \pm 0.04$ | $0.30 \pm 0.05$ | $0.89 \pm 0.01$ | $0.03 \pm 0.01$ | $0.53 \pm 0.02$ |
| AACL | $0.90 \pm 0.01$ | $-0.02 \pm 0.01$ | $0.57 \pm 0.02$ | $0.80 \pm 0.03$ | $0.04 \pm 0.03$ | $0.60 \pm 0.01$ |

Table 17: Continual learning metrics of ten methods in the **expansion** and **contraction** situations on three **Bigfish** tasks. The added baselines include TDGR, UPGD, and P&C. Results are reported as mean ± standard deviation over five seeds. The average continual return of ALL in Bigfish is 24.77, and in Atlantis is 296, 949, which are not provided in the table. The top three results are highlighted in green, and the depth of the color indicates the ranking.

| Methods | Expansion | | | Contraction | | |
|---|---|---|---|---|---|---|
| | Return↑ | Forgetting↓ | Transfer↑ | Return↑ | Forgetting↓ | Transfer↑ |
| IND | 13.86 ± 2.41 | −0.27 ± 0.08 | – | 1.66 ± 0.96 | 0.18 ± 0.02 | – |
| FT | 15.13 ± 0.10 | −0.25 ± 0.04 | 0.00 ± 0.02 | 3.01 ± 1.39 | 0.14 ± 0.08 | 0.23 ± 0.02 |
| EWC | 3.31 ± 1.88 | −0.00 ± 0.11 | −0.08 ± 0.12 | 1.74 ± 1.04 | 0.26 ± 0.06 | 0.16 ± 0.06 |
| online-EWC | 13.43 ± 2.02 | −0.31 ± 0.09 | 0.02 ± 0.05 | 1.84 ± 0.80 | 0.14 ± 0.03 | 0.18 ± 0.07 |
| Mask | 12.18 ± 1.45 | −0.19 ± 0.04 | −0.02 ± 0.03 | 1.49 ± 0.71 | −0.01 ± 0.01 | 0.06 ± 0.06 |
| CLEAR | 17.68 ± 4.79 | −0.04 ± 0.03 | 0.18 ± 0.07 | 1.48 ± 0.44 | 0.23 ± 0.02 | 0.11 ± 0.04 |
| P&C | 4.27 ± 1.98 | −0.05 ± 0.10 | 0.07 ± 0.04 | 4.67 ± 1.60 | 0.12 ± 0.08 | 0.07 ± 0.06 |
| TDGR | 17.33 ± 2.53 | −0.40 ± 0.03 | −0.06 ± 0.06 | 1.63 ± 2.55 | 0.26 ± 0.02 | 0.28 ± 0.02 |
| UPGD | 10.99 ± 2.80 | −0.23 ± 0.08 | −0.05 ± 0.07 | 4.89 ± 2.26 | 0.06 ± 0.07 | 0.10 ± 0.09 |
| AACL | 18.27 ± 1.22 | −0.05 ± 0.11 | 0.21 ± 0.05 | 10.03 ± 1.94 | 0.12 ± 0.07 | 0.19 ± 0.05 |

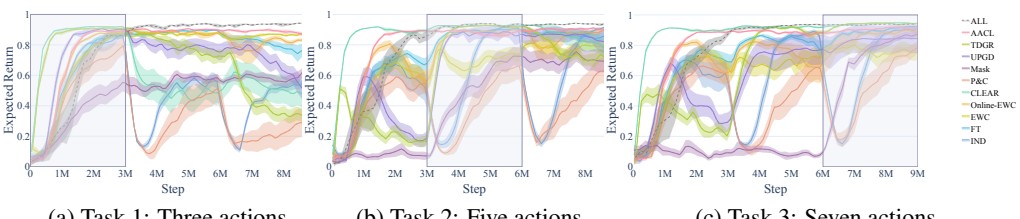

(a) Task 1: Three actions     (b) Task 2: Five actions     (c) Task 3: Seven actions

Figure 18: Performance of eleven methods on three **MiniGrid** tasks in the **expansion** situation.

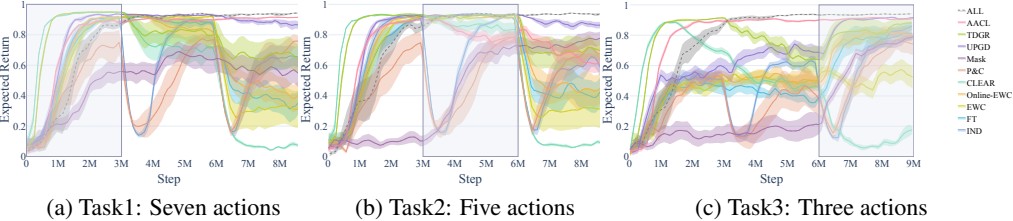

(a) Task1: Seven actions     (b) Task2: Five actions     (c) Task3: Three actions

Figure 19: Performance of eleven methods on three **MiniGrid** tasks in the **contraction** situation.

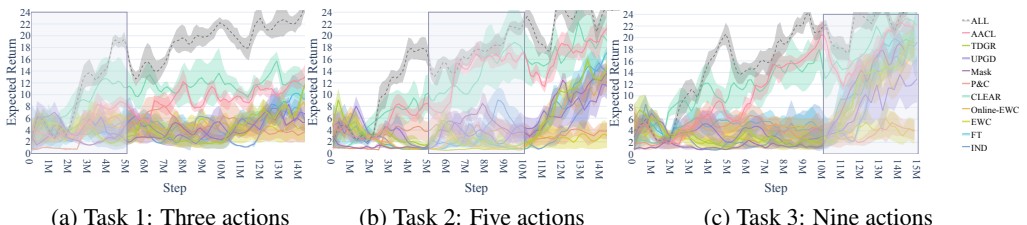

(a) Task 1: Three actions     (b) Task 2: Five actions     (c) Task 3: Nine actions

Figure 20: Performance of eleven methods on three **Bigfish** tasks in the **expansion** situation.

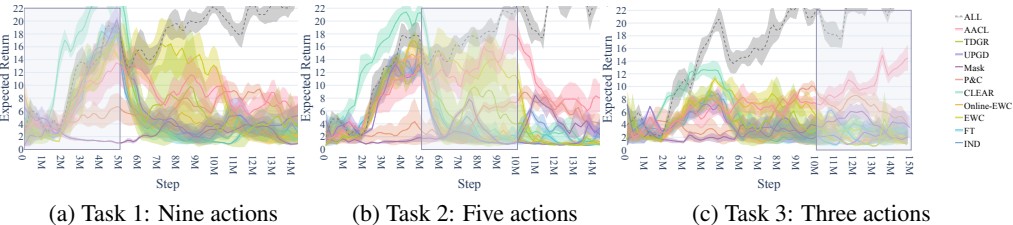

(a) Task 1: Nine actions     (b) Task 2: Five actions     (c) Task 3: Three actions

Figure 21: Performance of eleven methods on three **Bigfish** tasks in the **contraction** situation.

# E APPENDIX: PROBLEM DETAILS

## E.1 PROBLEM CONSTRAINTS

In this section, we elaborate on the assumptions and constraints imposed in the formulation of the Continual Learning with Dynamic Capabilities (CL-DC) problem, as introduced in the main text. The goal is to focus on the effect of dynamically changing action spaces on the CL process, while ensuring that the underlying "task logic" remains consistent across tasks. To this end, we formalize the following constraints:

**Reward Function Consistency:** For any pair of tasks $i$ and $j$ in the sequence, the reward functions $\mathcal{R}^i$ and $\mathcal{R}^j$ are required to be *conceptually similar*. Specifically, for any state transition $(S, S')$ and for any actions $A^i \in \mathcal{A}^i$, $A^j \in \mathcal{A}^j$ such that $A^i = A^j$, the following holds:

$$\mathcal{R}^i(S, S', A^i) = \mathcal{R}^j(S, S', A^j), \quad \text{if } A^i = A^j, \tag{10}$$

where $S, S' \in \mathcal{S}$, and $A^i$, $A^j$ denote actions in their respective action spaces. This constraint ensures that the reward structure associated with a particular action remains invariant across tasks, provided the action is present in both action spaces.

**Transition Function Consistency:** Similarly, the transition probability functions $\mathcal{P}^i$ and $\mathcal{P}^j$ are also required to be *conceptually similar*. For any $S, S' \in \mathcal{S}$ and actions $A^i \in \mathcal{A}^i$, $A^j \in \mathcal{A}^j$ such that $A^i = A^j$, we require:

$$\mathcal{P}^i(S' \mid S, A^i) = \mathcal{P}^j(S' \mid S, A^j), \quad \text{if } A^i = A^j. \tag{11}$$

This constraint guarantees that the environment dynamics associated with a specific action are preserved across different tasks whenever that action is available.

**Optimal Policy Consistency:** Under the above constraints on reward and transition functions, we further require that the optimal policy exhibits consistency across tasks for shared actions. Formally, we assume the initial state distributions of task $i$ and task $j$ are identical, then for any state $S \in \mathcal{S}$ and any action $A^i = A^j \in \mathcal{A}^i \cap \mathcal{A}^j$, the optimal policies satisfy:

$$\pi^{*i}(A^i \mid S) = \pi^{*j}(A^j \mid S). \tag{12}$$

This condition ensures that, for the overlapping portion of the action spaces, the optimal action-selection behavior remains unchanged across tasks. Note that our framework does not use optimal policy consistency for training or loss computation. It is a tool for fair benchmarking. Removing it would not invalidate empirical advantages, but would introduce other sources of variation.

The above constraints collectively ensure that the *task logic* remains unchanged for actions that are common across tasks. By enforcing these constraints, we are able to attribute any observed changes in learning performance or policy generalization solely to the variations in the action spaces, rather than to confounding changes in environment dynamics or reward structures.

## E.2 AN APPLICATION EXAMPLE

To illustrate the practical significance of CL-DC, we provide an example: Consider a cleaning robot navigating a 2D map. Initially, it can move in all directions to reach a target, earning a positive reward for reaching the target and incurring small negative rewards for each move. After learning, a hardware failure restricts its forward movement capabilities, reducing its action space. Despite this change, the transition dynamics for the same state-action pairs have not changed. For example, moving left still results in a leftward state transition. The rewards also remain consistent for the same actions. The robot must now adapt its policy to navigate to the target without direct forward movement. After the hardware is repaired, the robot's action space expands again. If the robot has CL ability to prevent catastrophic forgetting, it will quickly adapt to the previous action space and further retain its learned policy for future encounters with the same failure. This example illustrates the following concepts: 1) The underlying dynamics of the task do not fundamentally change. 2) The underlying dynamics of the task do not fundamentally change. 3) It is crucial to maintain performance in previous action spaces to ensure robustness and adaptability.

# F APPENDIX: MORE DISCUSSION

## F.1 CONNECTION WITH NEUROSCIENCE

The field of neuroscience offers valuable insights into the mechanisms underlying motor control and learning, which can inform the development of artificial intelligence systems, particularly in the context of CRL (Kaplanis et al., 2019; Gazzaniga et al., 2019; Kudithipudi et al., 2022).

In the human brain, motor control is distributed across several anatomical structures that operate hierarchically (D'Mello et al., 2020; Friedman & Robbins, 2022). At the highest levels, planning is concerned with how an action achieves an objective, while lower levels translate goals into specific movements. This hierarchical organization allows for flexible and adaptive behavior, as higher-level goals can be achieved through various lower-level actions depending on the context. Similarly, in AACL, the agent's policy can be seen as operating at a high level, focusing on achieving task objectives, while the action representation space operates at a lower level, translating these objectives into specific actions. By decoupling the policy from the specific action space, AACL leverages a hierarchical approach that mirrors the brain's strategy for motor control. This allows the agent to adapt to changes in the action space without needing to relearn the entire policy.

Neurophysiological studies have shown that the activity of neurons in the motor cortex is often correlated with movement direction rather than specific muscle activations (Georgopoulos & Pellizzer, 1995; Kakei et al., 1999). Neurons exhibit directional tuning, and their collective activity can be represented as a population vector that predicts movement direction. This concept of population coding suggests that the brain represents actions in a high-dimensional space, allowing for generalization across different contexts. In AACL, the action representation space serves a similar function. By encoding actions in a high-dimensional space, the agent can generalize its policy across different action spaces. The encoder-decoder architecture in AACL can be likened to the neural mechanisms that map cortical activity to specific movements. When the action space changes, the update of the encoder and the decoder, is akin to how the brain might update its motor representations in response to changes in the body or environment.

Recent research in motor neurophysiology has highlighted the dynamic nature of neural representations. Neurons do not have fixed roles but instead can represent different features depending on the context and time (Churchland et al., 2012; Gallego et al., 2020). This flexibility allows the motor system to adapt to a wide range of tasks and environments, providing maximum behavioral flexibility. AACL incorporates this idea by allowing the action representation space to be dynamically updated. This dynamic updating process is analogous to how the brain adjusts its neural representations to maintain consistent behavior despite changes in the body or environment.

By drawing inspiration from neuroscience, AACL achieve policy generalization and adaptability in the face of changing action spaces. This connection between neuroscience and artificial intelligence not only enhances our understanding of both fields but also provides feasible ideas for more sophisticated and adaptable AI systems.

## F.2 EXTENDING TO OTHER SITUATIONS

Our framework can be extended to other situations as it is not specifically designed for expansion and contraction situations. Both the partial change situation and the complete change situation can be viewed as simultaneous occurrences of expansion and contraction. In the partial change situation, some actions from the previous action space remain, while in the complete change situation, all previous actions are removed.

In the complete change situation, incorporating experience replay is a straightforward improvement of our framework. However, this approach incurs additional storage and computational costs. Thus, the trade-off between performance and efficiency is necessary. In addition, the method of generating pseudo samples (Shin et al., 2017; Yue et al., 2023) via representation space may be more suitable for practical privacy-preserving scenarios, where the agent cannot access the previously collected data.

For the partial change situation, the challenge lies in identifying the differences between action spaces. Our framework can be improved by integrating mechanisms to detect removed and newly

added actions, similar to novelty detection and class-incremental learning in the open-world setting (Masana et al., 2022; Sahisnu Mazumder, 2024; Li et al., 2024a). While clustering or classifying action representations can achieve this, it imposes higher demands on the self-supervised learning method. The action representations need to be well-separated in the representation space. Therefore, leveraging more information beyond state changes to learn action representations could be a valuable extension of our framework.

### F.3 LIMITATIONS AND FUTURE WORK

Despite the promising advancements of our work, several limitations remain to be addressed. A primary concern is the scalability of our framework when applied to large and complex action spaces, particularly in high-dimensional environments. Future avenues of research should focus on enhancing efficiency by developing more effective action representation learning algorithms.

In addition, our current work primarily focuses on discrete action spaces. However, continuous action spaces are prevalent in real-world applications. Dynamically changing continuous action spaces introduce significant challenges, including variations in both action dimensionality and action range. To the best of our knowledge, while a few reinforcement learning studies address discrete action space changes, research on continual learning with dynamic continuous action spaces is even more limited. Nevertheless, our definitions and framework could potentially be extended to continuous action spaces. For example, when the action space dimensionality changes, the types of changes can still be categorized analogously to our discrete case, such as addition or reduction of action dimensions. To accommodate this, the decoder in our framework would need to output values for each dimension, rather than probabilities for discrete actions. This adaptation may involve leveraging Gaussian distributions or other continuous representation learning techniques. Similarly, changes in the action range for each dimension could be handled by discretizing the continuous space or by adopting more sophisticated encoder-decoder architectures capable of modeling continuous outputs. In situations where both the dimensionality and range change simultaneously, a multi-granularity or hierarchical action representation may be required, where high-level structures capture dimensional changes and low-level components handle range adjustments. Developing such representations is a promising direction for future research.

Another limitation is that the current exploration policy in AACL may not suitable for more complex action spaces. Future work should explore advanced exploration strategies, potentially drawing from recent advances in curiosity-driven exploration (Pathak et al., 2017; Mazzaglia et al., 2022). Finally, we plan to extend our work to a broader range of CL settings, including both changes in the agent's internal capabilities and variations in the external environment. This may involve leveraging meta-learning strategies to improve generalization across a wider spectrum of capability evolutions and incorporating advanced transfer learning mechanisms to facilitate seamless knowledge integration from diverse environments and agent configurations.

