# OpenReview forum: "Action-Adaptive Continual Learning: Enabling Policy Generalization under Dynamic Action Space"
_ICLR.cc/2026/Conference — Submitted to ICLR 2026_

### Official Review · Reviewer_7ekJ · 2025-10-18

**Soundness:** 3
**Presentation:** 2
**Contribution:** 2
**Rating:** 4
**Confidence:** 4

**Summary:**

The paper introduces AACL, which decouples the policy from task-specific action spaces by learning an action representation space via SSL (encoder–decoder) and adaptively fine-tuning the decoder when the action space changes, balancing stability and plasticity.

**Strengths:**

The proposed AACL framework effectively decouples policy and action representations, achieving a better stability–plasticity trade-off and superior forward transfer on benchmarks such as MiniGrid.

Experimental results are comprehensive and reproducible, showing that AACL consistently outperforms replay-, regularization-, and architecture-based baselines in both expansion and contraction scenarios.

**Weaknesses:**

The approach currently supports only discrete and limited action spaces, assuming shared state spaces and similar task semantics, which restricts its applicability to continuous-control or heterogeneous domains.

In action-space contraction settings, AACL still experiences performance degradation and forgetting, indicating incomplete mitigation of catastrophic forgetting.

The method requires an exploration phase at the start of each task to collect transition data for self-supervised training, which limits its efficiency and practicality in strictly offline settings.

**Questions:**

1. In line 239, typo: "the auxiliary task of the action representation"
2. In line 287, why does the encoder need fine-tuning but does not require adding constraints? If the encoder only needs to be fine-tuned on new tasks, will the encoder quickly forget the state to encoder vector mapping relationships learned on historical tasks?
3. In Algorithm 1, the pseudocode does not show the sequential relationship of task training. The current pseudocode appears to randomly select and train across all tasks.
4. If the action space is contracted, does it require prior knowledge of the environment to perform masking? Can the method proposed in this paper only adapt to scenarios where environmental dynamics do not change significantly, because in this paper, the state space seems to be assumed unchanged by default, whereas in real-world scenarios the state space may change?
5. When the environment's action space is reduced, the action mask requires prior knowledge of the environment, so I believe the statement in line 344 is inappropriate.
6. In Figures 4 and 5, all methods need to be tested on all tasks during training, so do the curves in the figures represent the average test performance across all tasks? If each subfigure represents the model's performance on each corresponding subtask, then for Figure 6, we can clearly see that when AACL is trained to the third task, the first two tasks show significant performance degradation, which means the model experiences severe forgetting. However, in Table 2's Bigfish experiment, AACL's forgetting is very small. Why is this?
7. In Figures 16 and 17, why does the ALL algorithm exceed the return range shown in the figures? Does this mean that if the performance of all other algorithms were converted to normalized scores, they would show relatively poor performance? I suggest the authors use normalized performance as a metric, as this would better enable comparison of performance differences between different models.
8. The authors need to clarify which module parameters are shared and which module parameters are task-specific in the current method.
9. The current experiments contain task sequences of only three tasks, but in classic CRL tasks, such as CW20 and CW10, the task sequence lengths are 20 and 10 respectively. Therefore, I suggest the authors consider longer task sequences to fully demonstrate the effectiveness of the proposed method.
10. The method proposed in this paper is not applicable to situations where the state space changes, and state space changes frequently occur in real-world scenarios, such as changes in the number of robot sensors or robotic arm models. Furthermore, compared to the environments used in most current CRL tasks (Continual World), the tasks adopted in this paper are simpler, and the environmental dynamics changes are smaller. This can be seen from Figures 16 and 17, where we can observe that when training the first task, the model's performance on the second and third tasks also improves simultaneously, which means these three continual learning tasks are essentially almost identical.
11. Since the model parameters used by the current authors are not large, and the continual learning task sequence is very short, why not directly train a separate model for each task? In my view, this would not significantly increase storage and computational overhead, and parallel training could even further reduce physical time consumption.
12. This method has the potential of being applied to continuous high-dimensional CRL environments, so I suggest the authors compare with recent methods on classic tasks such as CW10 and CW20, for example L2M [1], t-DGR [2], Doya-DaYu[3], and UPGD [4].

[1] Learning to modulate pre-trained models in rl

[2] dgr: A trajectory-based deep generative replay method for continual learning in decision making

[3] Lifelong reinforcement learning via neuromodulation

[4] Addressing loss of plasticity and catastrophic forgetting in continual learning

**Details Of Ethics Concerns:**

nan

---

> ### Author Response · Authors · 2025-11-26
>
> Thank you for your thoughtful and detailed feedback, which has greatly helped us identify areas for improvement and clarification in our work. Please find our point-by-point responses below.
>
> ---
>
> ## W1. Limited to Discrete and Shared State Spaces
> Our current framework is primarily designed for discrete action spaces, assuming shared state spaces and similar task semantics. This was a **deliberate choice to isolate and study the challenges specific to dynamic action spaces**, while minimizing confounding factors from state space or reward dynamics changes. As discussed in Appendix F.2 (Limitations and Future Work), we acknowledge this limitation and have outlined the extension to continuous and heterogeneous domains as a key direction for future research.
>
> ---
>
> ## W2. Incomplete Mitigation of Forgetting in Contraction Settings
> We agree that catastrophic forgetting is not fully eliminated, especially in contraction scenarios. As noted in recent CRL literature, **achieving both strong forward transfer and zero forgetting without task-specific parameters remains a challenging open problem** [3,4]. Our approach strives to balance stability and plasticity, and although forgetting is reduced, it is not entirely avoided.
>
> ---
>
> ## W3. Exploration Phase and Offline Settings
> In practice, our method only requires a modest number of transitions ($10^4$ steps, see Table 5, Figure 13) collected via a random policy for each new task. This phase is brief (runtime comparisons in Appendix C.6) and can be replaced with pre-collected offline data, making the approach compatible with offline RL settings.
>
> ---
>
> ## Q1 and Q3.
> Thank you for catching these minor issues. We have revised the manuscript with blue text.
>
> ---
>
> ## Q2. Encoder Fine-Tuning and Forgetting
> 1. We chose not to regularize the encoder because its plasticity is crucial for learning a generalized policy. Our ablation studies show that **constraining the encoder leads to reduced forward transfer and overall performance** (AACL-E, Table 1). When learning action representation, constraining the decoder can already prevent the encoder from rapidly forgetting the mapping relationships learned in previous tasks.
> 2. In AACL, the policy is learned in a stable latent representation space, which represents the "intrinsic logic" of the task. **The risk of catastrophic forgetting lies in the decoder**. If the decoder undergoes drastic changes when adapting to a new action space, even if the strategy is stable, the behavior of its output (after passing through the decoder) will become inconsistent and suboptimal.
> 3. This design is analogous to CRL works that *regularize the actor but not the critic* [1].
>
> ---
>
> ## Q4 & Q5. Prior Knowledge and Action Masking in Contraction
> 1. Our method only needs to know which actions are invalid in the new action space to apply masking on the decoder outputs; no further environment-specific knowledge is required. As stated around line 323, other methods require prior knowledge of all tasks because they must know the maximal action space across unseen tasks, whereas our approach does not need to know the size of future action spaces in advance. We adapt the decoder structure only when encountering a new task. Therefore, the statement around line 344 is appropriate: **our method does not rely on prior knowledge beyond identifying action space currently**.
> 2. Our study focuses specifically on continual learning under changes in the action space rather than arbitrary changes. To ensure the problem is well-posed and to isolate the effect of action-space changes, we assume the state space remains unchanged and that environment dynamics are consistent on the intersection of action spaces (Appendix E). This design choice enables us to **analyze the challenge stemming purely from action-space changes**.
>
> ---
>
> ## Q6. Interpretation of Figures 4, 5, and 6; Forgetting Metric
> As noted around line 365, the curves in Figures 4, 5, and 6 are not averages across all tasks. Each subfigure reports the model’s **evaluation performance on its corresponding subtask** throughout the continual training process. In Figure 6, while AACL shows some performance decline on the first two tasks when training the third task, the overall forgetting remains relatively low. This is because:
> - The forgetting metric is computed on **normalized performance** (line 1041), making scores comparable across tasks and environments.
> - Forgetting is measured after the completion of subsequent tasks, aggregating the degradation induced by each later task (Eq. 8). For instance, after finishing Task 2, the induced forgetting on Task 1 is relatively small, which contributes to a low overall forgetting score despite later fluctuations.

---

> > ### Author Response · Authors · 2025-11-26
> >
> > ## Q7. Use of Normalized Performance and ALL Baseline
> > In more complex tasks with action-space contraction, continual learning becomes inherently harder. We include the ALL baseline to illustrate the challenge gap between changing and static action spaces (line 345), **not for direct comparison**.** Even on Task 1, ALL has access to the full, *unchanging action space and therefore can achieve higher returns than methods* that must adapt to evolving spaces. Normalizing all methods to the ALL baseline would then obscure the extent of forgetting and forward transfer caused by action-space changes, since ALL faces a **fundamentally different problem**. Our current evaluation thus avoids normalizing by ALL to preserve the interpretability of forgetting and transfer.
> >
> > ---
> >
> > ## Q8. Parameter Sharing
> > Our method does not use parameter isolation. The policy, encoder, and decoder parameters are **all shared across tasks**.
> >
> > ---
> >
> > ## Q9. Task Sequence Length
> > While longer sequences, such as CW20, provide comprehensive tests for general CRL, our work targets the distinct challenge of dynamic action spaces rather than the classic setting with static action spaces. Constructing long task sequences that isolate action-space changes while controlling for confounders (state dynamics and reward differences) is non-trivial and introduces additional benchmark design complexity. As an initial step, we focus on three-task sequences to ensure tight control over confounding variables. Nonetheless, we have conducted a longer sequence study with five tasks in Appendix D.4, which further supports the effectiveness of our method under extended sequences within our problem setting.
> >
> > ---
> >
> > ## Q10. State Space Changes and Task Simplicity
> > Our work deliberately controls for changes in state space, environment dynamics, and reward functions to focus on the challenges of dynamic action spaces (see line 173 and Appendix E). Addressing state space changes is indeed a valuable direction [2], but it is outside our current scope. The similarity of tasks in Figures 16 and 17 is by design, to isolate the action space factor.
> >
> > ---
> >
> > ## Q11. Training Separate Models per Task
> > Continual learning aims to achieve both knowledge retention and transfer. Training a separate model per task addresses forgetting but sacrifices forward transfer. Our IND baseline embodies this approach and indeed trains a standalone model for each task. Its performance on later tasks, when trained independently without leveraging prior knowledge, is not high—for example, in Figure 5, Task 3 during 6M–9M, and in Figure 6, Task 3 during 10M–15M (exponential moving average smoothing makes the curves not start from exactly zero). This demonstrates that while separate models may not drastically increase storage or training time in our small-scale setup, they fail to provide the cross-task knowledge reuse that continual learning methods like AACL are designed to capture.

---

> ### Author Response · Authors · 2025-11-26
>
> ## Q12. Comparison with CW10/CW20 and Recent Methods
> Thank you for the suggestion to compare against recent CRL methods on CW10/CW20. We agree that these are strong and widely used baselines. At the same time, we would like to emphasize that **our problem setting fundamentally differs from CW10/CW20**: our focus is on continual learning under dynamic action spaces with controlled environment dynamics and reward functions. In contrast, CW10/CW20 assume static action spaces while emphasizing changes in dynamics/rewards. As a result, **direct comparison is not entirely fair for any method**, because the sources of non-stationarity are different and require different algorithmic inductive biases.
>
> Nevertheless, **we conducted additional experiments to bridge this gap as much as possible**. Among the four methods you suggested:
> - Doya-DaYu is not yet published, and no official code is available.
> - L2M targets the adaptation of large, pre-trained models and is tailored to a pre-train-then-finetune paradigm, which is orthogonal to our setting without large-scale pretraining.
>
> Therefore, we compare against the other two available methods (**t-DGR** and **UPGD**) and additionally include **P&C** $[^5]$, a hybrid method combining structural and regularization-based ideas.
>
> The new results on MiniGrid are summarized below.
>
> |             |                 | Expansion       |               |                 | Contraction     |               |
> | ----------- | --------------- | --------------- | ------------- | --------------- | --------------- | ------------- |
> | Methods     | Return↑         | Forgetting↓     | Transfer↑     | Return↑         | Forgetting↓     | Transfer↑     |
> | TDGR        | 0.66 ± 0.03     | 0.33 ± 0.03     | 0.30 ± 0.02   | 0.74 ± 0.17     | 0.18 ± 0.06     | 0.47 ± 0.04   |
> | UPDG        | 0.73 ± 0.08     | 0.17 ± 0.04     | 0.30 ± 0.05   | **0.89 ± 0.01**     | **0.03 ± 0.01**     | 0.53 ± 0.02   |
> | P&C         | 0.59 ± 0.08     | 0.17 ± 0.08     | 0.36 ± 0.05   | 0.69 ± 0.07     | 0.10 ± 0.05     | 0.40 ± 0.07   |
> | AACL        | **0.90 ± 0.01**     | **-0.02 ± 0.01**    | **0.57 ± 0.02**   | 0.80 ± 0.03     | 0.04 ± 0.03     | **0.60 ± 0.01**   |
>
>
> The new results on Bigish are summarized below.
>
> |             |                 | Expansion       |               |                 | Contraction     |               |
> | ----------- | --------------- | --------------- | ------------- | --------------- | --------------- | ------------- |
> | Methods     | Return↑         | Forgetting↓     | Transfer↑     | Return↑         | Forgetting↓     | Transfer↑     |
> | TDGR    | 17.33±2.53 | **-0.40±0.03** | -0.06±0.06 | 1.63±2.55 | 0.26±0.02 | **0.28±0.02** |
> | UPDG    | 10.99±2.80 | -0.23±0.08 | -0.05±0.07 | 4.89±2.26 | **0.06±0.07** | 0.10±0.09 |
> | P&C     | 4.27±1.98  | -0.05±0.10 | 0.07±0.04  | 4.67±1.60 | 0.12±0.08 | 0.07±0.06 |
> | AACL    | **18.27±1.22** | -0.05±0.11 | **0.21±0.05** | **10.03±1.94** | 0.12±0.07 | 0.19±0.05 |
>
>
> Key observations:
> - Overall, **AACL maintains the strongest performance among all methods**. In the Expansion setting, it achieves *higher continual returns and higher forward transfer*; in the Contraction setting on Bigish tasks, it continues to deliver the *best continual return*.
> - These findings align with our original conclusions: *action-space generalization via the action representation space is particularly beneficial for zero-shot transfer to new action spaces*. While still challenging in the contraction situation, AACL’s forward transfer and overall average performance are consistently strong.
>
> We are running additional experiments on Atlantis. Due to longer experimental times, we will continue updating the added results in **Appendix D.10** to keep the main text concise for the time being.
>
> Thank you again for your constructive feedback and suggestions, which will help us further improve the clarity and impact of our work.
>
> ---
>
> **References:**
> [1] Disentangling transfer in continual reinforcement learning. NeurIPS, 2022.
> [2] Multi-granularity Knowledge Transfer for Continual Reinforcement Learning. IJCAI, 2025.
> [3] Humans and neural networks show similar patterns of transfer and interference during continual learning. Nat Hum Behav, 2025.
> [4] Principled Fast and Meta Knowledge Learners for Continual Reinforcement Learning. ICLR, 2026.
>
> [5] Progress & Compress: A Scalable Framework for Continual Learning, ICML, 2018.

---

### Official Review · Reviewer_RTpF · 2025-10-19

**Soundness:** 2
**Presentation:** 2
**Contribution:** 2
**Rating:** 6
**Confidence:** 2

**Summary:**

The paper proposes a tokenized, transformer-based world model for continual reinforcement learning (CRL). It introduces a compact VQ-VAE tokenizer to discretize frames, a causal Transformer world model (WM) trained for next-step token/reward/done prediction, and an actor–critic policy trained on imagined rollouts. Adaptation is localized using LoRA adapters inserted only into the world model, while a FiLM-based task conditioning module fuses explicit task IDs with trajectory-inferred context.

**Strengths:**

# Strengths
* The use of tokenized (VQ-VAE) state representations in a world model tailored for CRL is innovative and bridges discrete representation learning with continual adaptation.
* Adapting only the world model while freezing the policy is a well-motivated design choice to stabilize learning and isolate plasticity.
* The study includes detailed metrics (Isolated Forgetting, Forward Transfer) on six Atari tasks with clear ablations and competitive baselines.
* The paper is technically detailed, with well-structured methodology, hyperparameter listings, and pseudo-code. It’s easy to reproduce and interpret.

**Weaknesses:**

# Weaknesses
* Evaluation is confined to Atari CORA; no experiments on non-visual or real-world-like domains (e.g., DM Control or robotic settings).
* Dependency on Task Cues
* It remains somewhat unclear whether improvements arise mainly from LoRA-based modularity or from the discrete tokenization (the ablation focuses only on LoRA).
* Some tables are a bit nonsensical (mainly Table 1. Why do you need a table that has a single row?)

**Questions:**

# Questions / Suggestions
* Including a runtime or compute comparison to baselines would strengthen the practical relevance.
* How does the tokenizer quality (e.g., codebook size or reconstruction error) affect CRL performance?

---

> ### Author Response · Authors · 2025-11-26
>
> Thank you for your feedback. We appreciate your time and effort in reviewing our work and offer the following point-by-point responses.
>
> ## W1: Scope of Evaluation and Choice of Domains
> Our current experiments focus on discrete action spaces and do not include DM Control or robotic settings. As an exploratory step, our work targets discrete actions to establish the core principles of policy generalization under dynamic action spaces. To ensure diversity and challenge, we evaluate across three distinct environments with different task sequences and action-space dynamics (expansion, contraction, and mixed). Expanding to non-visual and continuous-control domains is part of our planned future work, and we have outlined this direction in the Limitations and Future Work section, including the need for more sophisticated representation learning and exploration strategies tailored to continuous action spaces.
>
> ## W2: Dependency on Task Cues
> Our method does not require task cues or task IDs, but instead relies on task boundaries. This assumption aligns with common continual reinforcement learning practice, where task transitions are acknowledged but explicit task labels are avoided. Similar to standard reinforcement learning setups, the agent also needs to know the current task’s action space.
>
> ## W3: Source of Improvement and Use of LoRA
> We do not use LoRA in our framework. All modules are trained and updated across tasks with shared parameters, and improvements stem from decoupling the policy from the action space via the action representation space.
>
> ## W4: Table Formatting and Single-Row Table
> Table 1 presents continual learning metrics of eight methods and three variants of AACL across three MiniGrid tasks in situations of expansion and contraction. It demonstrates that AACL consistently outperforms other baselines.
>
> ## A1: Runtime Comparison
> We provide wall-clock runtime comparisons in Appendix C.6, measured under identical hardware and training protocols. The results indicate that AACL incurs a moderate overhead compared to standard baselines due to the additional modules and exploration, but remains computationally efficient and scalable for practical use.
>
> ## A2: Tokenizer Quality and Its Impact
> Our method does not rely on a discrete tokenizer. The action representation is learned via an encoder-decoder trained with self-supervised objectives on collected transitions, and adaptation proceeds by fine-tuning when the action space changes. Therefore, codebook size or reconstruction error in a tokenizer is not applicable to our framework.

---

### Official Review · Reviewer_9teN · 2025-10-21

**Soundness:** 2
**Presentation:** 1
**Contribution:** 2
**Rating:** 2
**Confidence:** 3

**Summary:**

This paper investigates the problem of continual learning (CL) under dynamic capability, where the agent must handle changing action spaces across tasks. To address this, the authors propose the Action-Adaptive Continual Learning (AACL) framework, which decouples the policy network from task-specific action spaces by introducing a shared action representation space. When encountering a new action space, AACL fine-tunes the encoder–decoder components to adapt the representation accordingly. Furthermore, the authors release a benchmark designed to evaluate this setting and demonstrate that AACL achieves superior performance compared to existing methods.

**Strengths:**

1.	The paper tackles an underexplored and practically relevant problem—continual learning with dynamically changing action spaces—thereby extending the conventional CL setting beyond fixed-action environments.
2.	Experimental results across the proposed benchmark indicate that AACL achieves consistently strong performance, validating the effectiveness of the action representation mechanism.

**Weaknesses:**

1.	The paper should more explicitly articulate the fundamental challenges of continual learning with dynamic capability (CL-DC) compared to standard CL. Is the key difficulty solely the expansion or contraction of the action dimension, or does it introduce deeper representational conflicts? Moreover, how does this setting relate to class-incremental CL, where the output head expands over time?
2.	It remains unclear how the model determines whether the action space has expanded or contracted. Does the framework assume access to a task identifier during training or inference? If the encoder or decoder dynamically changes its structure, how is inference performed without explicit knowledge of the current task or action configuration?
3.	Although regularization is used to mitigate catastrophic forgetting in the decoder, continuous fine-tuning may still alter the mapping from latent representations to action distributions. When the action space expands or contracts, previously learned action distributions may shift undesirably. The paper should explain how AACL preserves decoding consistency for earlier tasks under such structural transitions.
4.	The selection of baselines appears somewhat dated, with the most recent comparison being Mask (2023). Considering that the paper targets ICLR 2026, a fair evaluation would require including more recent continual RL methods. Furthermore, as AACL combines structure-based (encoder) and regularization-based (decoder) strategies, both categories (also combination) should be represented among the baselines to ensure methodological fairness.
5.	The ablation analysis focuses solely on regularization components, while neglecting structural variations such as applying structured expansion or mask to the decoder. Additionally, the current analysis provides limited insight into the causal factors underlying the observed results; deeper interpretation of why certain components contribute more substantially would enhance the paper’s scientific rigor.

**Questions:**

See the weakness.

---

> ### Author Response · Authors · 2025-11-26
>
> Thank you for your thoughtful and constructive feedback. We appreciate your time and effort in reviewing our work and offer the following point-by-point responses.
>
> ## W1: Challenges of CL-DC
> We appreciate the opportunity to clarify the unique challenges of Continual Learning with Dynamic Capabilities (CL-DC). Unlike standard CL, where agents adapt to environmental changes *assuming a static action space*, CL-DC introduces **two synergistic challenges**:
>
> - **Action Space Dynamics**: The core difficulty lies in **policy generalization across action spaces of varying sizes**, which involves both expansion (e.g., hardware upgrades) and contraction (e.g., damage), not just dimension scaling. This is distinct from class-incremental CL, where output heads expand monotonically without contraction. CL-DC’s bidirectional changes disrupt the policy-action mapping, causing representational conflicts where optimal actions for past tasks become invalid or suboptimal under new spaces.
> - **Task Logic Preservation**: Crucially, CL-DC requires maintaining **consistent task logic** (identical state transitions and rewards for shared actions) across varying action spaces. This imposes deeper representational conflicts: policies must generalize despite shifting action distributions (Figure 2), where the *optimal policy* may fundamentally differ (e.g., loss of actions in Atlantis ).
>
> Our benchmark experiments highlight this: methods like CLEAR and EWC fail catastrophically during contraction, as they lack mechanisms to reconcile action-space-induced policy shifts. This is unlike class-incremental CL, which solely addresses output-head growth.
>
> ---
>
> ## W2: Action space determination
> AACL **does not require explicit task identifiers** during training or inference. Instead, it dynamically adapts action-space changes via structural updates to the decoder:
>
> - **Expansion**: New neurons are added to the decoder output layer, with weights randomly initialized while existing weights remain fixed.
> - **Contraction**: Outputs for removed actions are masked.
>     The decoder’s structure is updated **on-the-fly** upon detecting changes in the action space size, which is inferred from the **current environment's action availability**.
>
> **Inference** relies solely on the current decoder configuration:
>
> - The encoder maps state transitions to latent action representations.
> - The decoder translates these representations to action probabilities **for the current action space**. Actions that are currently unavailable in the environment will not be effective.
> No prior task knowledge is needed; the decoder adapts structurally, while the policy operates on the stable latent space.
>
> ---
>
> ## W3: Decoder Consistency under Structural Changes
> AACL ensures decoding consistency through **elastic weight consolidation (EWC)** applied exclusively to the decoder:
>
> - **Regularization During Fine-Tuning**: The decoder parameters are constrained using Fisher information matrices from prior tasks to minimize drift in latent-action mappings.
> - **Encoder Plasticity**: The encoder remains *unregularized* to adaptively refine the latent space, avoiding rigidity.
>
> **Empirical evidence** validates this design:
>
> - The variant **AACL-O** (no decoder regularization) shows higher forgetting and lower forward transfer (Table 1), confirming EWC’s role in preserving decoding consistency.
> - **AACL-E** (encoder regularized) performs worse in forward transfer, proving encoder plasticity is essential for adapting representations.
> - **Action Space Stability**: Figure 14 demonstrates that latent representations of shared actions (e.g., "turn left") remain clustered across tasks, while new actions embed naturally. Regularization ensures minimal disruption to existing mappings.
>
> Thus, while fine-tuning may slightly shift distributions, EWC guarantees that latent representations for **shared actions** remain stable, enabling seamless generalization across structural transitions.

---

> ### Author Response · Authors · 2025-11-26
>
> ## W4: Baseline Selection
> Our problem setting differs fundamentally from the dominant CRL literature that primarily targets environmental non-stationarity (e.g., dynamics/reward drift) under a fixed action space. Consequently, many recent CRL methods are not directly applicable, or their assumptions render comparisons methodologically misaligned.
> However, in order to better reflect the challenges encountered by these methods in CL-DC problems, our work still compares three typical CL methods in CL, including one replay-based method, two regularization-based methods, and one architecture-based method.
> Therefore, due to the unique nature of CL-DC, “apples-to-apples” fairness is inherently limited.
>
> Nevertheless, **we conducted additional experiments to bridge this gap as much as possible**. We compare against **P&C**$[^3]$ (a classical hybrid method combining structural and regularization-based ideas) and two other recent CRL methods (**t-DGR** $[^1]$ and **UPGD** $[^2]$).
> The new results on MiniGrid are summarized below.
>
> |             |                 | Expansion       |               |                 | Contraction     |               |
> | ----------- | --------------- | --------------- | ------------- | --------------- | --------------- | ------------- |
> | Methods     | Return↑         | Forgetting↓     | Transfer↑     | Return↑         | Forgetting↓     | Transfer↑     |
> | TDGR        | 0.66 ± 0.03     | 0.33 ± 0.03     | 0.30 ± 0.02   | 0.74 ± 0.17     | 0.18 ± 0.06     | 0.47 ± 0.04   |
> | UPDG        | 0.73 ± 0.08     | 0.17 ± 0.04     | 0.30 ± 0.05   | **0.89 ± 0.01**     | **0.03 ± 0.01**     | 0.53 ± 0.02   |
> | P&C         | 0.59 ± 0.08     | 0.17 ± 0.08     | 0.36 ± 0.05   | 0.69 ± 0.07     | 0.10 ± 0.05     | 0.40 ± 0.07   |
> | AACL        | **0.90 ± 0.01**     | **-0.02 ± 0.01**    | **0.57 ± 0.02**   | 0.80 ± 0.03     | 0.04 ± 0.03     | **0.60 ± 0.01**   |
>
>
> The new results on Bigish are summarized below.
>
> |             |                 | Expansion       |               |                 | Contraction     |               |
> | ----------- | --------------- | --------------- | ------------- | --------------- | --------------- | ------------- |
> | Methods     | Return↑         | Forgetting↓     | Transfer↑     | Return↑         | Forgetting↓     | Transfer↑     |
> | TDGR    | 17.33±2.53 | **-0.40±0.03** | -0.06±0.06 | 1.63±2.55 | 0.26±0.02 | **0.28±0.02** |
> | UPDG    | 10.99±2.80 | -0.23±0.08 | -0.05±0.07 | 4.89±2.26 | **0.06±0.07** | 0.10±0.09 |
> | P&C     | 4.27±1.98  | -0.05±0.10 | 0.07±0.04  | 4.67±1.60 | 0.12±0.08 | 0.07±0.06 |
> | AACL    | **18.27±1.22** | -0.05±0.11 | **0.21±0.05** | **10.03±1.94** | 0.12±0.07 | 0.19±0.05 |
>
>
> Key observations:
> - Overall, **AACL maintains the strongest performance among all methods**. In the Expansion setting, it achieves *higher continual returns and higher forward transfer*; in the Contraction setting on Bigish tasks, it continues to deliver the *best continual return*.
> - These findings align with our original conclusions: *action-space generalization via the action representation space is particularly beneficial for zero-shot transfer to new action spaces*. While still challenging in the contraction situation, AACL’s forward transfer and overall average performance are consistently strong.
>
> We are running additional experiments on Atlantis. Due to longer experimental times, we will continue updating the added results in **Appendix D.10** to keep the main text concise for the time being.
>
> ---
>
> ## W5: Additional Ablations
> Structural changes to the decoder are integral, rather than optional, components of AACL. Without adaptive expansion/contraction, the agent cannot produce valid probabilities for newly added actions or properly retire removed ones. In other words, “structured expansion” (or masking) is a prerequisite for feasibility under CL-DC, not merely a variant to ablate away. That is why our ablations concentrated on regularization choices, which are the main drivers of the stability–plasticity trade-off given a required structural update.
>
> [1] t-DGR: A trajectory-based deep generative replay method for continual learning in decision making, CoLLAs, 2025.
>
> [2] Addressing loss of plasticity and catastrophic forgetting in continual learning, ICLR, 2024.
>
> [3] Progress & Compress: A Scalable Framework for Continual Learning, ICML, 2018

---

### Official Review · Reviewer_NSfQ · 2025-11-01

**Soundness:** 2
**Presentation:** 2
**Contribution:** 2
**Rating:** 2
**Confidence:** 3

**Summary:**

This paper introduces Continual Learning with Dynamic Capabilities (CL-DC), a problem where agents must learn sequentially while their action spaces dynamically change through expansion, contraction, partial change, or complete change. The authors propose Action-Adaptive Continual Learning (AACL), which decouples policies from specific action spaces by learning an action representation space via self-supervised learning (SSL). An encoder maps actions to representations, while a decoder maps representations to action probabilities. When action spaces change, the decoder structure is adaptively updated and both encoder-decoder are fine-tuned with EWC-based regularization. The authors release a benchmark based on MiniGrid, Procgen (Bigfish), and ALE (Atlantis) environments. Experiments show AACL outperforms baselines including FT, EWC, Online-EWC, CLEAR, and Mask on expansion and contraction scenarios.

**Strengths:**

### **1. Problem formulation**:
CL-DC is a practical and underexplored problem in continual RL with clear real-world motivation like hardware failures and software updates.

### **2. Consistent improvements**:
AACL outperforms a comprehensive set of baselines across most metrics and environments.

**Weaknesses:**

### **1. Limited Technical Novelty**
The approach essentially combines existing techniques (SSL for representation learning + EWC regularization + adaptive architecture). The core contribution is incremental rather than introducing fundamentally new methods. The SSL objective (predicting actions from state transitions) is adapted from prior work (Chandak et al., 2019; Fang & Stachenfeld, 2024), and EWC is a standard continual learning technique.

### **2. Doubtful Scalability**
The authors only tested their approach where actions are discrete and with very low dimensionality (maximum 9 actions). It also requires the agents to explore whenever action spaces change in order to fine-tune the action encoder-decoder. I believe this can work in simple environments as tested by the authors, but in an environment with complex dynamics or in real-world settings, the sample efficiency and applicability remain unknown. Specific concerns include:

- Fisher information matrix storage scales as O(|parameters|²), raising scalability doubts
- Random exploration policy acknowledged as potentially insufficient for complex spaces
- No evaluation on continuous action spaces, critical for real robotic applications

It would be better if the authors provide further discussion and empirical evidence addressing these scalability concerns.

### **3. Catastrophic Forgetting's Significance Not Justified**
The authors propose avoiding catastrophic forgetting as one of their main novelties, but its importance needs further justification. Two examples are given to reason the need to adapt to a dynamic action space: hardware/software updates in robotic systems, and structural change during the life of a biological system. In the latter example, a swift and stable adaptation is needed, but it never requires the agents to maintain good performance in the exact previous forms.
For robotics, I also don't see the need to maintain the performance of previous versions in the same policy network. On the other hand, if generalizing across different action spaces for same tasks is the goal, there is a whole line of research on cross-embodiment learning that fits this case better, including:
- **"RoboCat: A Self-Improving Robotic Agent" (Bousmalis et al., 2023)** - learns action representations across multiple robot embodiments and action spaces
- **"Scaling Cross-Embodied Learning: One Policy for Manipulation, Navigation, Locomotion and Aviation"** (Doshi et al., 2024) - learns abstract action representations that transfer across embodiments

### 4. **Overly Simple Experiment Design for a Benchmark**
The experimental design has only discrete actions and there is only one task for each environment. More critically, the authors define 4 types of action space changes (Section 3.2): Expansion, Contraction, Partial Change, and Complete Change, but only expansion and contraction are tested in the main experiments. Appendix D.3 shows sequential expansion-then-contraction, which is **not** the same as true partial change. There is no further justification or discussion other than to punt it to future work in Appendix G.1.

### 5. **Clarity in Writing**
- The first sentence of the abstract says "Continual Learning (CL) is a powerful tool..." but continual learning is a research topic or problem setting, not a "tool." Tools would be specific methods or algorithms designed for continual learning.
- The authors repeatedly use the overloaded word "capability" to refer to action space, which can be confusing. Simply using "action space" would be clearer.
- Page 5, line 239: "Therefore, the auxiliary task **off** the action representation" should be "of".

**Questions:**

Can you please address the concerns over novelty, scalability, limitations in experiment design, and why mitigating catastrophic forgetting is essential in this particular continual learning setting?

---

> ### Author Response · Authors · 2025-11-26
>
> Thank you for your thoughtful and detailed review. We appreciate your constructive feedback and the opportunity to clarify our contributions, assumptions, and future directions.
>
> ## W1. Limited Technical Novelty
> Our contribution is not only a mere combination of existing techniques. The central novelty lies in formulating and addressing **Continual Learning with Dynamic Capabilities (CL-DC)**, **a new setting** in which the agent’s action space itself changes over time. This is a dimension largely ignored in prior continual learning work that assumes static capabilities.
>
> Our framework is designed around a neuroscience-inspired decoupling: the policy is trained in a latent action representation space, while the decoder adaptively maps that representation to the current action space. This architectural separation—policy stability in the latent space and plasticity in the decoder—follows cortical-cerebellar functional roles and grounds how we integrate components:
> - **Self-supervised learning**: We design an SSL objective that predicts actions from state transitions to construct an action representation space tied to environment dynamics.
> - **Adaptive architecture**: The decoder is structurally adapted to expansion and contraction of the action space via targeted addition/removal or masking of output units.
> - **Selective EWC**: EWC is applied only to the decoder to protect previously learned representation-to-action mappings without impeding encoder plasticity. Our ablations (AACL-O) show significant performance drops without decoder regularization.
>
> Thus, **the novelty is not in isolated techniques but in their principled orchestration to solve CL-DC**. We also release a benchmark across three environments tailored to this setting. We view this as a **paradigm-level contribution** that introduces and operationalizes a new continual learning problem.
>
> ---
>
> ## W2. Doubtful Scalability
> We acknowledge that this work is a **first step toward CL-DC** and have discussed limitations and future directions in the manuscript. Nonetheless, we respond to your specific concerns:
>
> - **Fisher matrix storage**: In AACL, EWC is applied only to the decoder, which is a small portion of parameters. For example, in Bigfish/Atlantis, the decoder is a single linear layer (256 to 7), making O(|θ|²) storage manageable. Moreover, efficient EWC variants (e.g., Online EWC) can be incorporated to further reduce memory.
> - **Exploration efficiency**: Random exploration suffices under our problem definition and environments. Ablations (**Appendix D.7**) show that using the previous task’s policy for exploration introduces bias and can degrade performance. Sensitivity analysis (**Appendix D.5**) indicates that only a modest number of steps (e.g., 5,000) yields good results, and excessive exploration may harm generalization. For more complex environments, we agree that advanced exploration (e.g., curiosity-driven strategies) is a promising direction.
> - **Continuous action spaces**: We intentionally focused on discrete actions to isolate the impact of capability changes. Appendix F (Limitations) outlines extensions to continuous actions and more sophisticated representation learning. Our benchmark is designed to seed future work in these more complex settings.
>
> As an additional scalability check, we conducted a preliminary test on Minigrid with a longer task sequence. Building on the setup in Appendix D.4, we repeated the five tasks four times to form **a sequence of length 20**. We compared AACL with Online-EWC. Results over three random seeds are shown below:
>
> | Methods     | Return↑       | Forgetting↓   | Transfer↑     |
> |--          |--            |--            |--            |
> | Online-EWC  | 0.73±0.21     | 0.03±0.02     | 0.29±0.03     |
> | AACL        | 0.80±0.19     | 0.03±0.07     | 0.35±0.00     |
>
> These results suggest that, on longer sequences, both methods experience metric degradation may be due to plasticity loss. Nevertheless, **AACL maintains clear advantages in sustained return and positive transfer**. Given the longer runtime, we will continue to expand Appendix D.4 with more seeds and additional baselines.

---

> ### Author Response · Authors · 2025-11-26
>
> ## W3. Catastrophic Forgetting's Significance Not Justified
> Our objective is not to retain perfect performance for all past action spaces within a single policy simultaneously. Rather, we aim to **maintain a stable policy over the task’s core logic in the latent space** and prevent catastrophic forgetting in the decoder, the component responsible for translating latent intents into concrete actions. If the decoder’s previously learned mappings are erased during adaptation, the same latent policy can yield inconsistent or suboptimal behaviors across capability changes.
>
> Consider a robot undergoing a hardware update from differential-drive to omnidirectional motion. The core policy (e.g., path planning toward goals) need not change; the decoder must remap latent intentions to new motor commands. If later the robot reverts to its older actuation (e.g., maintenance or compatibility), the ability to switch back reliably depends on retaining prior decoder mappings. **We illustrate this scenario in Appendix E.2**. In short, robustness across capability changes necessitates guarding against decoder-level forgetting, even when task logic is unchanged.
>
> Regarding cross-embodiment learning (e.g., RoboCat; Doshi et al.), we see CL-DC as complementary but distinct. Cross-embodiment typically spans different MDPs with different morphology, sensors, and dynamics across multiple agents, whereas CL-DC focuses on a single agent whose capabilities evolve **over time** under the same task logic and shared state space. Our goal is temporal adaptability for one embodied agent, rather than training a universal policy across heterogeneous embodiments. Thus, CL-DC addresses a new axis of continual learning.
>
> ---
>
> ## W4. Overly Simple Experiment Design for a Benchmark
> We deliberately **isolate the variable of action space change** under a single task logic with discrete actions to cleanly study the CL-DC phenomenon. This design clarifies the mechanism of representation-policy decoupling and decoder adaptation without confounding factors from task shifts or continuous control optimization.
>
> - Partial change via sequences: Our sequences (expansion→contraction and contraction→expansion) test the core capabilities required for partial change, which can be viewed as combinations of adding and removing actions. Sequentially composing these operations provides a **systematic** way to probe stability-plasticity trade-offs more deeply than a single-step random partial change.
> - Complete change: We agree that complete change (disjoint action spaces) is challenging and important. Appendix G.1 discusses why aligning representation spaces without shared actions is difficult and outlines approaches like experience replay or generative models. We refrained from including these in the main experiments to **keep focus on the overlapping-action regimes most aligned with our framework’s assumptions and to avoid conflating distinct technical challenges**. We see complete change as a substantial follow-on topic.
>
> We believe our benchmark provides a **clear, controlled starting point** for the community and demonstrates AACL’s effectiveness on dynamic action spaces.
>
> ---
>
> ## W5. Clarity in Writing
> Thank you for the constructive writing suggestions. We have revised the manuscript.
>
> We appreciate your feedback and believe the clarifications and revisions strengthen the paper’s presentation and positioning.

---

### Meta-Review · Area_Chair_RcyM · 2026-01-02

**Summary:**

This paper proposes a new continual reinforcement learning setting that assumes the action space changes over time while the state space remains fixed. The paper is well organized and clearly written. However, it does not introduce a novel method, instead relying on existing continual learning techniques to mitigate forgetting, resulting in limited technical contribution and only modest improvements in alleviating forgetting. In addition, the experimental evaluation considers only discrete action space scenarios, while ignoring more realistic settings with continuous action spaces and changing state spaces, which limits the applicability and significance of the work.

**Reviewer Concerns:**

Reviewer RTpF’s comments do not align with the content of the manuscript. Reviewer NSfQ and Reviewer 7ekJ both argue that restricting the action space to be discrete is unrealistic for practical robotic applications. In addition, Reviewer 7ekJ points out that the paper ignores realistic scenarios in which the state space also changes. Reviewer NSfQ considers the technical contribution of the paper to be limited, as it mainly combines existing continual learning methods. These issues were not adequately addressed during the rebuttal period.

**Reviewer Scores:**

Reviewer NSfQ believes that the paper proposes a good setting. During the rebuttal, the authors clarified the four settings described in the method section and demonstrated the effectiveness of the approach with a larger action space (20 discrete actions). However, regarding the reviewer’s main concerns about the technical contribution of the paper—namely, that the work is merely a combination of existing methods (SSL+EWC) without introducing a novel approach, and that the evaluation is limited to discrete action spaces with a small number of actions rather than continuous action spaces—the authors’ responses did not fully address these concerns.

Reviewer 9teN asked the authors to clarify the challenges of the newly proposed Continual Learning with Dynamic Capabilities setting and how the action space is determined, which the authors did. In addition, the authors added 2–3 new baselines. The reviewer is likely to increase the score from 2 to 4.

Reviewer RTpF’s review is largely unrelated to the content of the paper.

Reviewer 7ekJ argues that the paper only considers settings with a fixed state space and discrete action spaces, which restricts its applicability to continuous-control or heterogeneous domains. Moreover, the proposed method does not account for changes in the state space. In addition, the data requirements of the proposed approach limit its effectiveness and practicality in strictly discrete environments. Finally, the reviewer raised more than ten specific questions, which the authors addressed during the rebuttal. However, the limitations related to discrete action spaces and dynamically changing environments are still not adequately handled or empirically validated.

---

### Decision · Program_Chairs · 2026-01-26

Reject